# Emotional tones of voice affect the acoustics and perception of Mandarin tones

**Hui-Shan Chang**[1,2,3]**, Chao-Yang Lee**[2,4]**, Xianhui Wang**[4]**, Shuenn-Tsong Young**[5]**, Cheng-Hsuan Li**[3]**, Woei-Chyn Chu**[1]*****

1 Department of Biomedical Engineering, National Yang Ming Chiao Tung University, Taipei City, Taiwan,
2 Department of Audiology and Speech-Language Pathology, Asia University, Taichung City, Taiwan,
3 Graduate Institute of Educational Information and Measurement, National Taichung University of Education, Taichung City, Taiwan, 4 Division of Communication Sciences and Disorders, Ohio University, Athens, Ohio, United States of America, 5 Institute of Geriatric Welfare Technology and Science, MacKay Medical College, New Taipei City, Taiwan

* wchu@nycu.edu.tw

**Data Availability Statement:** All relevant data are within the manuscript and its Supporting Information files.

**Funding:** This study was supported by grants MOST 109-2218-E-010-003 and MOST 110-2622-

## Abstract

Lexical tones and emotions are conveyed by a similar set of acoustic parameters; therefore, listeners of tonal languages face the challenge of processing lexical tones and emotions in the acoustic signal concurrently. This study examined how emotions affect the acoustics and perception of Mandarin tones. In Experiment 1, Mandarin tones were produced by professional actors with angry, fear, happy, sad, and neutral tones of voice. Acoustic analyses on mean F0, F0 range, mean amplitude, and duration were conducted on syllables excised from a carrier phrase. The results showed that emotions affect Mandarin tone acoustics to different degrees depending on specific Mandarin tones and specific emotions. In Experiment 2, selected syllables from Experiment 1 were presented in isolation or in context. Listeners were asked to identify the Mandarin tones and emotions of the syllables. The results showed that emotions affect Mandarin tone identification to a greater extent than Mandarin tones affect emotion recognition. Both Mandarin tones and emotions were identified more accurately in syllables presented with the carrier phrase, but the carrier phrase affected Mandarin tone identification and emotion recognition to different degrees. These findings suggest that lexical tones and emotions interact in complex but systematic ways.

## Introduction

Speech conveys more than the linguistic message intended by a speaker. It provides information about the speaker such as physical characteristics, regional accent, and emotional state. Since multiple sources of information often converge on the same acoustic parameters, the two fundamental questions are how the different sources of information contribute to speech acoustics, and how listeners disentangle these sources of information during speech perception. In this study, we investigated the relationship between emotional tones of voice (emotions hereafter) and lexical tones by examining how four common emotions shape the acoustic characteristics of Mandarin tones, and how the emotions affect the perception of Mandarin tones.

B-A49-001 from the Ministry of Science and Technology, Taiwan ROC. The funders had no role in study design, data collection and analysis, decision to publish, or preparation of the manuscript.

**Competing interests:** The authors have declared that no competing interests exist.

Emotional tone is defined as the vocal expression of emotion, which conveys a speaker's affective states, motivational states, or intended emotions [1–5]. The primary acoustic correlates of vocal emotions include fundamental frequency (F0), mean amplitude, and duration [1, 2, 6]. Previous research showed that F0 is the primary acoustic correlate of emotions [7–10], whereas amplitude and duration serve as secondary cues [11, 12]. Importantly, F0 and amplitude are highly correlated with each other [8].

Lexical tones are used to distinguish words in tonal languages. In Mandarin, segmentally identical words can be distinguished on the basis of F0 height or contour. For example, the syllable /ba/ means "eight", "uproot", "grip", or "father" with Tone 1 (a high-flat tone), Tone 2 (mid-rising), Tone 3 (mid-falling-rising), or Tone 4 (high-falling), respectively. The primary acoustic correlate of lexical tones is F0 [13]. Amplitude and duration also vary systematically among Mandarin tones [13–15], and both contribute to Mandarin tone perception as secondary cues [16–19]. However, F0 remains the most powerful cue for the perception of Mandarin tones [19–21].

Since the acoustic characteristics most relevant for lexical tones coincide with those for emotions, the convergence raises the question of how emotions affect the acoustics and perception of lexical tones. A tonal language like Mandarin offers a unique opportunity to examine this question.

## Theories of emotion

The two approaches to the analysis of emotion are the dimensional theory of emotion and the theory of basic emotions [22]. The difference between these two approaches is that emotions are either described as independent dimensions [23] or discrete entities [24]. In the dimensional approach, Russell (1980) [23] proposed a circlex model of emotion, which showed that each emotion could be arranged in a circle controlled by two orthogonal dimensions in space: valence and arousal [25–28]. The position of each emotion on the quadrant reflects different amounts of valence and arousal traits [27, 29]. The valence dimension is associated with a person's subjective feeling, ranging from displeasure to pleasure. The arousal dimension is associated with the energy of a person's subjective feeling, ranging from sleep to excitement [28].

The theory of basic emotions suggests that human emotions are composed of a limited number of basic emotions [30]. Each basic emotion has its proprietary neural circuits which are structurally different [24, 25, 31, 32]. Although the idea of basic emotions is commonly accepted, there is no consensus on the exact number of basic emotions. Plutchik (1962) [33] proposed eight primary emotions (anger, fear, sadness, disgust, surprise, anticipation, trust, and joy). Ekman (1992) [24, 34] proposed seven basic emotions (fear, anger, joy, sad, contempt, disgust, and surprise), but later changed to six (happiness, anger, sadness, fear, disgust, and surprise). Izard [35] proposed seven basic emotions (fear, anger, happiness, sadness, disgust, interest, and contempt). Recent studies examining facial expressions, neural mechanisms, and brain imaging suggest that the number of basic emotions could be further reduced to four (fear, anger, joy, and sadness) [36–40]. As an exploratory study of the tone-emotion relationship, we adopt the framework of four basic emotions (anger, fear, happiness, and sadness) in the current study.

## How emotions affect speech acoustics

There is ample evidence that different emotions result in distinct acoustic characteristics [2, 5–7, 41–54]. Physiologically, the sympathetic nervous system is aroused by emotions such as anger, fear, or happiness, resulting in a higher heart rate and blood pressure, a dry mouth, and occasionally muscle tremors [55, 56]. Consequently, speech is loud, fast, and has intense high-

frequency energy. On the other hand, sadness arouses the parasympathetic nervous system. Heart rate and blood pressure decrease, salivation increases, and speech is produced slowly and with little high-frequency energy. These physiological changes are reflected in amplitude, energy distribution across the frequency spectrum, frequency of pauses, and duration. For example, higher arousal associated with excitement, fear, and anger have been shown to generate higher mean F0 [7, 9, 57], higher mean amplitude [58–61], and shorter duration [56].

The influence of specific emotions on speech acoustics varies across studies. When compared with a neutral tone of voice, an angry voice has a higher mean F0, a wider or similar F0 range, a higher mean amplitude, and a shorter duration [2, 5–7, 41–51, 54]. A fearful voice shows a higher mean F0, a narrower, wider, or similar F0 range, a higher or lower mean amplitude, and a shorter duration [2, 5–7, 41–51, 54]. A happy voice has a higher mean F0, a wider or similar F0 range, a higher or equal mean amplitude, and a shorter or longer duration [2, 5–7, 41–54]. A sad voice has a lower or similar mean F0, a narrower or wider or similar F0 range, a lower mean amplitude, and a longer duration [2, 5–7, 41–54]. In sum, some emotions have fairly consistent acoustic features, whereas other emotions are more variable. The variability, however, is consistent with the idea that emotion is sociocultural in nature, i.e., there are cross-linguistic and cross-cultural differences in the acoustic manifestation of emotions [62]. The variability is also consistent with the observation that features of emotions vary across speakers, sexes, and contexts [63].

Several studies compared the acoustic characteristics of emotions between tonal and non-tonal languages. Ross, Edmondson, and Seibert (1986) [64] examined acoustic characteristics of neutral, happy, sad, angry, and surprising emotions using Thai, Taiwanese, Mandarin (all tonal languages), and English (a non-tonal language). They found greater F0 variations in English compared to the tonal languages, suggesting that non-tonal languages have a greater degree of freedom in using F0 to convey emotions. In contrast, no significant difference was found in duration or amplitude between the tonal and non-tonal languages.

Anolli, Wang, Mantovani, and De Toni (2008) [65] investigated acoustic differences among happy, sad, angry, fear, scornful, prideful, guilty, and shameful emotions in Mandarin and Italian (a non-tonal language). They found that emotions were characterized by significant variations in F0 and amplitude for Italian but not for Mandarin. In contrast, duration varied significantly among the emotions for Mandarin but not for Italian. Since Italian is a syllable-timed language, it is also likely that the less variation of syllable duration in Italian reflects the language-specific prosodic structure.

Wang, Lee, and Ma (2016, 2018) [46, 66] examined acoustic correlates of angry, fear, happy, sad, and neutral emotions in Mandarin and English. Semantically-neutral declarative sentences were embedded in different contexts to elicit angry, fear, happy, sad, and neutral emotions. Comparable English sentences were constructed with a direct translation of the Mandarin sentences. Acoustic analysis showed that F0 variations among the emotions were significantly greater in English than in Mandarin. In contrast, duration variations were significantly greater in Mandarin than in English.

Studies using other tonal languages also show more restricted F0 variations for emotions in a tonal language (Chong, Kim, and Davis, 2015 [67] for Cantonese; Luksaneeyanawin, 1998 [68] for Thai). To our knowledge, the only exception to this pattern is Li, Jia, Fang, and Dang's (2013) [69], who showed greater F0 variations associated with emotions in Mandarin compared to Japanese, which is a non-tonal language that uses lexical pitch accent extensively.

Regarding the effect of emotions on specific lexical tones, Chao (1933) [70] noted that Mandarin uses successive additional tones and edge tones to implemet the intonation for emotions (see Liang and Chen, 2019 [71], for an illustration). Li, Fang, and Dang (2011) [44] examined how emotions affect the F0 and duration of Mandarin utterances ranging from one to fourteen

syllables. The results showed that anger and disgust were associated with an additional falling tone, and happiness and surprise were associated with an additional rising tone. Non-neutral emotions resulted in a different F0 range, register, contour, or duration. For example, happiness and surprise were associated with a higher F0 range and higher register, whereas sadness and disgust were associated with a reduced F0 range and lower register.

In sum, most studies comparing tonal and non-tonal languages show that F0 variations associated with emotions are greater in non-tonal languages. This suggests that lexical tones constrain the availability of F0 for emotions in tonal languages. In contrast, amplitude or duration variations associated with emotions appear to be greater in tonal languages [14, 64, 65], suggesting that amplitude or duration may be used to compensate for the restricted use of F0 in conveying emotions. Studies on Mandarin further showed that emotions shape F0 and duration characteristics of Mandarin tones.

## How emotions affect speech perception

The speech perception literature shows that emotions affect speech perception at various levels of processing. Mullennix, Bihon, Bricklemyer, Gaston, and Keener (2002) [1] examined how variations in emotions and talker voice affect spoken word recognition in English. They presented pairs of names (e.g., *Todd-Tom*) produced by either the same or different talkers, and with the same or different emotions. The participants' task was to judge whether the names in a pair were the same or different. The results showed that variations in emotion slowed down judgments of both the names and talker voices, indicating emotion affected perception of consonants and talker characteristics. Kitayama and Ishii (2002) [72] and Ishii et al. (2003) [73] presented words spoken in a pleasant or unpleasant tone of voice. While ignoring the emotional tone, listeners were asked to judge whether the word meaning was pleasant (e.g., *grateful*, *satisfaction*) or unpleasant (e.g., *complaint*, *dislike*). The results showed that emotion variations slowed down judgments of word meaning. Nygaard and Lunders (2002) [52] examined how emotions (happy, neutral, and sad) affect the perception of homophonic words (e.g., *die/dye*). They found that selection of word meaning was compromised by the emotion of the words. Nygaard and Queen (2008) [53] presented happy (e.g., *cheer*), sad (e.g., *upset*), or neutral (e.g., *chair*) words spoken with consistent, inconsistent, or neutral emotions. Listeners were asked to repeat the words they heard. The results showed that listeners responded more quickly when the meaning of the words matched the emotions.

Similarly, research on tonal languages show the impact of emotions on speech perception, and the effect is further modulated by tonal language experience. Singh, Lee, and Goh (2016) [74] examined how changes in emotion and Mandarin tone affect consonant recognition, and how consonant changes affect emotion recognition and Mandarin tone identification. For Mandarin-speaking listeners, variations in Mandarin tone and emotion made consonant recognition less accurate. Consonant variations also made Mandarin tone identification and emotion recognition less accurate. Consonants and prosody (Mandarin tone and emotion) affect each other to the same extent. In contrast, for English-speaking listeners, consonant recognition was affected by prosodic variation to a different degree than prosody recognition was affected by consonant variations. That is, the effects of emotions and lexical tones on segmental perception depend on tonal language experience.

Liang and Chen (2019) [71] examined how emotions and tonal language experience affect Mandarin tone perception. Four Chinese pseudo words (i.e., *mong*, *ging*, *ra*, *bü*) were created, and each had four lexical tones variations. Each syllable-tone combination was embedded in the middle of carrier phrases. The syllable immediately before the pseudo words was manipulated to create four tonal contexts (i.e., *chu1*, *du2*, *xie3*, *lian4*). For instance, *mong1* was

embedded in carrier phrases (1) *zhi3 **chu1** [mong1] zhe4ge0 zi4* "Please point out the word [*mong1*]"; (2) *wo3 hui4 **du2** [mong1] zhe4 ge4 zi4*" I can read the word [*mong1*]"); (3) *wo3 hui4 **xie3** [mong1] zhe4 ge4 zi4*" I can write the word [*mong1*]"; (4) *wo3 xiang3 **lian4** [mong1] zhe4 ge4 zi4*" I can practice the word [*mong1*].*" All stimuli were produced with an angry, happy, sad, or neutral emotion. Mandarin listeners and Dutch-speaking learners of Mandarin were asked to identify the Mandarin tone of the pseudo words. The results showed that stimuli produced with the neutral emotion resulted in higher accuracy than those produced with non-neutral emotions. However, only the angry voice resulted in significantly lower accuracy relative to the neutral voice for both groups of listeners. In addition, Tone 4 was identified more accurately than Tone 1 in the angry voice.

In sum, research on speech perception shows that emotions affect the perception of segmental phonemes, talker voices, and lexical tones. The effect of emotion on lexical tone perception depends on both stimulus characteristics and tonal language experience. Particularly relevant to the current study, Liang and Chen's (2019) [71] findings further demonstrated that emotions affect lexical tone perception to different degrees depending on specific Mandarin tones.

## How emotions are perceived in speech

In addition to understanding emotion's effect on speech perception, we also explore how emotions themselves are perceived in speech. A common approach to studying emotion recognition is to recruit professional actors to produce speech materials with different emotions. Listeners are then asked to identify the emotions of the stimuli [2, 4, 7, 50, 75–77]. Studies using non-tonal languages showed that sad and angry voices are easier to identify than fearful and happy voices [2, 4, 7, 50, 75–77]. Studies using Mandarin have reported similar findings: negative emotions such as sadness [41, 46, 66] and fear [43] are easier to identify than positive emotions such as happiness [41, 43]. It has been suggested that negative emotions are prioritized in vocal communication because they convey warnings in situations of attack, loss, and danger. Consequently, negative emotions need to be communicated more effectively to ensure human survival [43, 78, 79]. In contrast, positive emotions such as happiness are usually expressed through additional communication channels (e.g., facial expression), which may explain why a happy voice is identified with lower accuracy when only the vocal channel is used [7, 43, 50].

There is limited evidence on how lexical tones affect the perception of emotions. Wang, Ding, & Gu (2012) [80] investigated emotion recognition from Mandarin sentences by native and non-native speakers. Mandarin words with various tones (*qi4che1* "car", *zhao4pian4* "picture", *xin1fang2* "new house", *dian4nao3* "computer", and *xue2xiao3* "school") were embedded in a semantically neutral carrier phrase (*zhe4 shi4 ta1 de0 [target word]* "This is his [target word]"). The sentences were recorded with six emotions (happiness, fear, anger, sadness, boredom, & neutral) and presented to listeners for emotion recognition. The results showed that native listeners had an overall higher accuracy than non-native listeners, but both groups recognized sadness with the highest accuracy and boredom with the lowest accuracy. Since tones were not systematically manipulated, it is not clear whether the results could inform the effect of lexical tone on emotion recognition.

Wang and Lee (2015) [41] and Wang and Qian (2018) [47] constructed sentences composed exclusively of a particular Mandarin tone (e.g., *wang1 bin1 xing1qi1tian1 xiu1 fei1ji1* "Wang Bin fixed the airplane on Sunday" or with a mixture of different tones (e.g., *wo3 bu4gan3 xiang1xin4 zhe4 shi4 zhen1de0* "I cannot believe this is true"). The sentences were recorded with various emotions. It was hypothesized that emotions would be recognized less

accurately in the Tone 1-only sentences because of the restricted F0 variation imposed by the (level) tone. An alternative hypothesis was that the restricted F0 variation in the Tone 1 sentences would have allowed emotions to surface more easily, thus facilitating emotion recognition. However, the results showed that emotions were recognized equally well regardless of tonal composition; that is, the restricted F0 variation imposed by the level tone did not compromise or facilitate emotion recognition. Emotional recognition seems quite robust irrespective of specific lexical tones.

## Benefit of context

The presence of context can alter a listener's interpretation of a speech sound [81]. Such perceptual adaptation forms the basis of speaker normalization [82, 83]. In speech audiometry, a carrier phrase is typically included in a word recognition task to provide a cue for the listeners to focus their attention on the target words [84–86]. Previous studies showed that word recognition accuracy is typically higher when embedded in a carrier phrase [84, 85, 87, 88]. The presence of a carrier phrase is particularly helpful under challenging listening conditions. For example, Lynn and Brotman (1981) [85] found that word identification in a carrier phrase was 10% more accurate than in isolation in the presence of speech-shaped noise. Since lexical tones produced with emotions are likely to deviate from the citation form, the presence of a context is likely to help listeners retrieve the intended tones more effectively.

In the current study, we examine Mandarin tone identification and emotion recognition in two contexts: when the target syllables are embedded in a carrier phrase (in context), and when the target syllables are extracted from the carrier phrase (in isolation). Note that the syllables in the "isolation" condition were not produced in isolation; rather, they were excised from the carrier phrase. We predict that Mandarin tone identification would be less accurate when the target syllables were extracted from the carrier phrase. This is because the citation form of a tone is likely to be altered due to the influence of neighboring tones [89]. Without the carrier phrase, it would be challenging for listeners to recover the tone. Furthermore, the carrier phrase provides information about talker characteristics such as speaking F0 range, which has been shown to facilitate tone identification from the multi-talker input [90]. We also predict that emotion recognition would be less accurate when the target syllables are presented in isolation. Since the talkers who recorded the stimuli were instructed to produce emotions for the entire utterance, the presence of emotions when preceded by a carrier phrase should facilitate emotion recognition from the target syllables.

## The present study

The above review shows that both acoustics and perception of speech are shaped by emotions. Emotions affect Mandarin tone identification to different degrees depending on specific tones [71], but specific Mandarin tones do not seem to affect emotion recognition differently [41, 47]. To further clarify the interaction between lexical tones and emotions in speech, this study examines the acoustics and perception of Mandarin tones produced with various emotions, and the perception of emotions embedded in Mandarin tones. Following Liang and Chen (2019) [71], we use syllables produced with four Mandarin tones in a sentence-medial position. Extending Liang and Chen (2019) [71], we use multiple speakers of both sexes to record the stimuli. We also examine both the acoustics and perception of Mandarin tones and emotions. Finally, we examine the perception of Mandarin tones and emotions in two contexts: when the target syllables are presented with the carrier phrase, and when the target syllables are extracted from the carrier phrase.

Based on prior research, we predict that the acoustics of emotions would be affected by specific Mandarin tones. Liang and Chen's (2019) [71] findings lead us to predict that the accuracy of Mandarin tone identification would be affected by emotions to different degrees depending on specific emotions and Mandarin tones in the stimuli. Following findings from Wang and Lee (2015) [41] and Wang and Qian (2018) [47], we predict that emotion recognition would remain robust irrespective of the specific Mandarin tones in the stimuli. Finally, we predict that Mandarin tone identification and emotion recognition would be less accurate when the target words are extracted from the carrier phrase.

## Experiment 1

In this experiment, we investigated the acoustic characteristics of Mandarin tones produced with emotions by multiple talkers of both sexes. Anger (ANGRY hereafter), fear (FEAR hereafter), happiness (HAPPY hereafter), and sadness (SAD hereafter) were selected because they are considered four basic emotions [36–40] (see introduction for opposing views). The acoustic effects of these four emotions were evaluated relative to the neutral tone of voice (NEUTRAL hereafter). Four acoustic measures including mean F0, F0 range, amplitude, and duration were chosen because they are most relevant to both lexical tone and emotional tone distinctions.

Based on the literature reviewed, we expect that ANGRY would result in a higher mean F0, a wider or similar F0 range, a higher mean amplitude, and a shorter duration when compared to the NEUTRAL baseline. FEAR would result in a higher mean F0, a narrower, wider, or similar F0 range, a higher or lower mean amplitude, and a shorter duration. HAPPY would result in a higher mean F0, a wider or similar F0 range, a higher or equal mean amplitude, and a shorter or longer duration. SAD would result in a lower or similar mean F0, a narrower or wider or similar F0 range, a lower mean amplitude, and a longer duration. These predictions are summarized in Table 1. We also expect that the acoustic differences among the emotions would be modulated by specific Mandarin tones.

### Method

**Talkers.** The use of human subjects in this study was reviewed and approved by the Institutional Review Board of National Yang Ming Chiao Tung University (IRB No. 1000063). Written informed consent was obtained from all talkers. No minors participated in this study. No medical records or archived samples were used in this study.

Eight professional actors (4 women and 4 men; mean age of 32.4 ± 7.1 years) were recruited to record the speech materials. All were native speakers of Taiwan Mandarin with no reported history of speech, hearing, or language disorders. Each talker was compensated $1,600 New Taiwan Dollars ($54 USD) per hour for their participation.

**Speech materials.** Three syllables /fa/, /Ji/, and /pHu/ with the four Mandarin tones were selected as target syllables, resulting in 12 syllable-tone combinations [發/fa1/], [筏/fa2/], [髮/

**Table 1. Summary of predicted acoustic characteristics of the four emotions relative to the neutral emotion in Experiment 1.** The symbols >, <, and = indicate an emotion is associated with a higher, lower, or comparable value compared to the neutral emotion.

|  | Mean F0 | F0 range | Mean amplitude | Duration |
|---|---|---|---|---|
| **Angry** | > | > or = | > | < |
| **Fear** | > | > or = or < | > or < | < |
| **Happy** | > | > or = | > or = | > or < |
| **Sad** | < or = | > or = or < | < | > |

fa3]/, [法/fa4/], [西/Ji1/], [錫/Ji2/], [洗/Ji3/], [夕/Ji4/], [鋪/pHu1/], [葡/pHu2/], [譜/pHu3/], and [瀑/pHu4/]. These syllables were chosen because: (1) they included the three most common vowels in Taiwan Mandarin [91, 92], (2) all began with a voiceless or aspirated consonant to facilitate identification of syllable onset, (3) all syllable-tone combinations were real words in Mandarin, and (4) the meanings of all syllable-tone combinations were emotionally neutral.

The 12 syllable-tone combinations were paired with five emotions (ANGRY, FEAR, HAPPY, SAD, and NEUTRAL) and embedded in a semantically neutral carrier phrase /*ni3 ʂuo1 [target word] ts5*/ "You say the word [target word]". In the carrier phrase, the syllable following the target syllable began with a voiceless consonant /*ts5*/ to facilitate identification of syllable boundaries. The 60 word-emotion combinations were produced twice by eight talkers for a total of 960 stimuli.

**Procedure.** Speech recordings took place in a sound-treated booth in the Department of Biomedical Engineering of National Yang Ming Chiao Tung University with a GRAS Type 40AC microphone at 0-degree azimuth. The microphone was placed 30 centimeters from the participant's mouth. The sampling rate was 44,100 Hz with 16-bit quantization. Before the recording, the first author discussed with the actors the emotional tones that they should aim for. The actors then completed the recording in the booth while being monitored by the first author.

The 120 stimuli were recorded in five blocks separated by emotions. The order of the blocks and the order of the stimuli within a block were randomized for each participant. Before the recording started, the participants were given 10 minutes to familiarize themselves with the stimuli. Breaks were given between blocks for the participants to adjust their emotions. Our goal was to elicit a broad-focus analysis, i.e., distributing the prosodic change over the whole sentence instead of focusing narrowly on the target syllable. To that end, the participants were instructed to avoid pausing before and after the target syllables, and to avoid placing excessive emphasis on the target syllables.

**Acoustic and statistical analysis.** The recordings (except for the NEUTRAL stimuli) were rated by 30 native speakers of Taiwan Mandarin to evaluate how well the intended emotions were present in the speech materials. The raters included 19 women and 11 men with ages ranging from 21 to 49 years (mean age 33.5 ± 7.8 years). For each stimulus, the raters were asked in a four-alternative forced-choice task to choose an emotion (ANGRY, FEAR, HAPPY, or SAD) that best represented the speech sample they heard. The raters were also asked to provide a score on a Likert's 5-point scale indicating the degree of the match, with 1 being the worst match and 5 being the best match. Stimuli were chosen for acoustic analyses only if they were correctly identified by all 30 participants and if they received an average rating of 3.0 and above. All stimuli met both criteria and were included in the acoustic analyses.

The acoustic analyses were performed with the Praat program [93]. Two landmarks were identified from the waveform: (1) the last glottal pulse of the syllable immediately before the target syllable, and (2) the last glottal pulse of the target syllable. The target syllable was then extracted based on these two landmarks. The acoustics measures were taken from the target syllables including mean F0, F0 range, mean amplitude, and duration.

## Results

Fig 1 shows the F0 contours of the four Mandarin tones produced with the five emotions. The F0 contours were time-normalized and averaged over speakers of the same sex. Specifically, for each token, F0 was measured every 10% from the beginning (0%) to the end (100%) to obtain 11 data points. Each of these 11 points was then averaged over all speakers of the same sex.

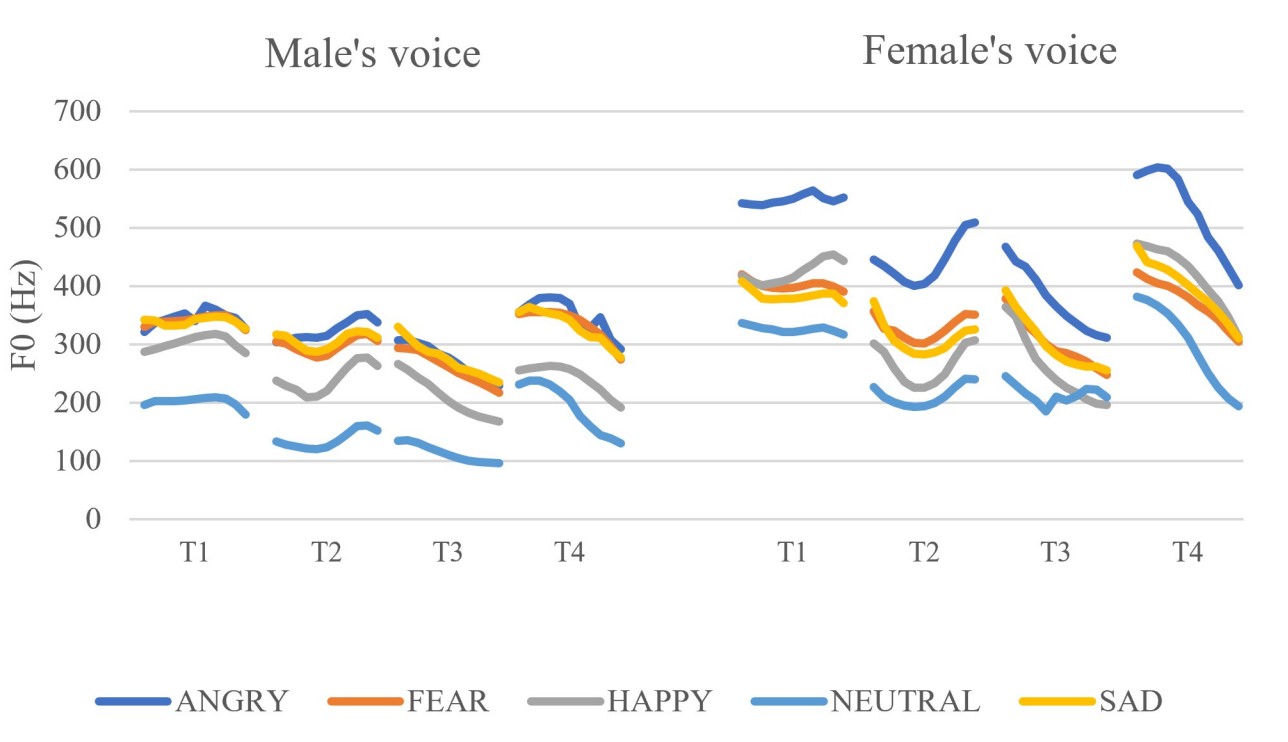

**Fig 1. F0 contours of target syllables as a function of emotion, Mandarin tone, and talker sex.**

The F0 contours of the Mandarin tones appear to be consistent with traditional descriptions of the Mandarin tones in citation form: Tone 1 is flat, Tone 2 is falling then rising, Tone 3 (which is in a non-final position in the carrier phrase) is falling, and Tone 4 is falling but in a higher register. The F0 plot is meant to show that the Mandarin tones were produced as intended. For quantitative analysis of the F0 contours, a more rigorous approach such as Functional Data Analysis should be taken [94, 95].

To evaluate the acoustic difference between the emotions, for each of the four acoustic measures (mean F0, F0 range, mean amplitude, and duration), a linear mixed-effects model was built for each of the four measures separately using R 3.6.3 (R Core Team, 2021) [96]. Mandarin tone (T1, T2, T3, and T4), emotion (ANGRY, FEAR, HAPPY, SAD, and NEUTRAL), and the tone-emotion interaction were entered as fixed effects. Talker, talker sex, syllable type, and repetition were entered as random effects.

**Mean F0.** Fig 2 shows the mean F0 of the target syllables as a function of emotion and Mandarin tone. Fig 1 suggests that the overall F0 contours of the Mandarin tones produced with the four emotions are similar to those of NEUTRAL; therefore, we calculated mean F0 as a summary measure for quantitative comparisons. The linear mixed-effects model revealed significant main effects of Mandarin tone, $\chi^2(3, N = 8) = 973.95, p < .001$, emotion, $\chi^2(4, N = 8) = 1749.66, p < .001$, and tone-emotion interaction, $\chi^2(12, N = 8) = 70.18, p < .001$. Post hoc pairwise comparisons (Tukey adjusted) were conducted to disentangle the interaction (Fig 2). It was found that ANGRY had the highest mean F0 and NEUTRAL had the lowest. The ranking of FEAR, HAPPY, and SAD varies on specific Mandarin tones. Full output of the model is available in S1 Table.

**F0 range.** Fig 3 shows the F0 range of the target syllables as a function of emotion and Mandarin tone. The linear mixed-effects model revealed significant main effects of Mandarin

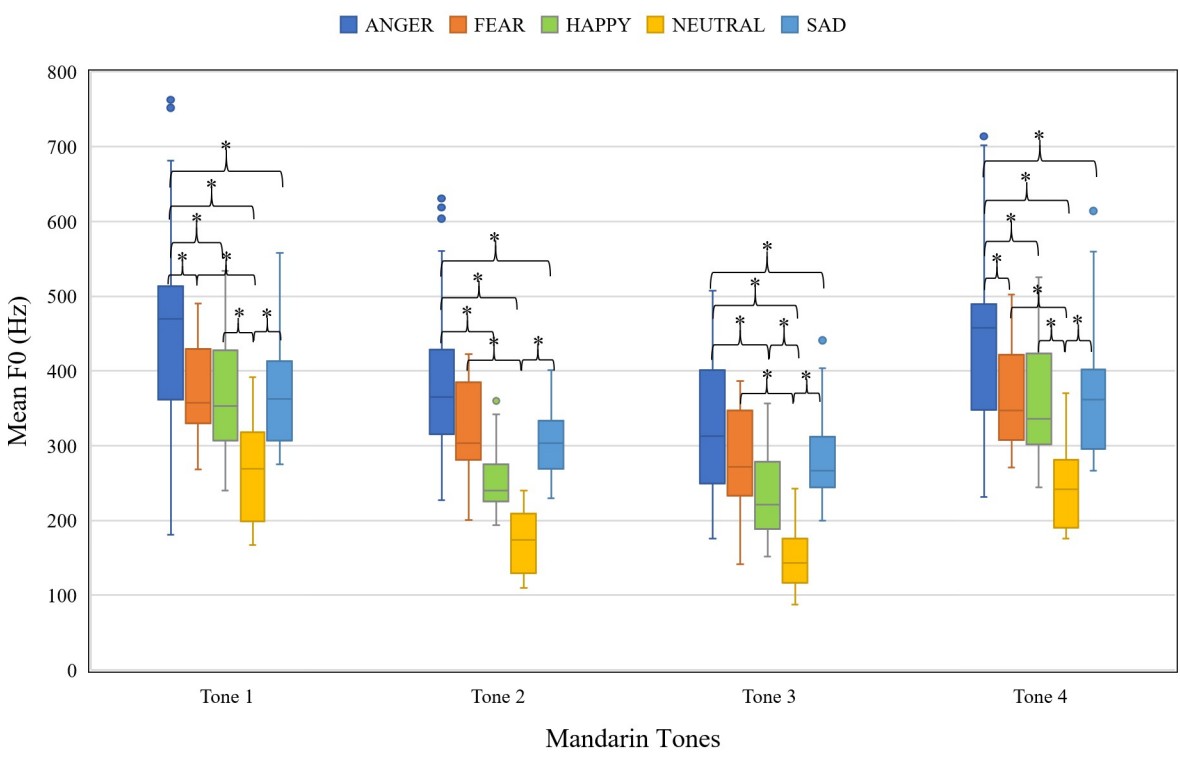

**Fig 2. Boxplot showing the mean F0 of target syllables as a function of Mandarin tone and emotion.** ($^*p < .05$).

tone, $\chi^2(3, N = 8) = 779.23$, $p < .001$, emotion, $\chi^2(4, N = 8) = 123.02$, $p < .001$, and their two-way interaction, $\chi^2(12, N = 8) = 114.64$, $p < .001$. The results of post hoc pairwise comparisons are also shown on the figure. There does not appear to be a consistent pattern in the ranking of the emotions. Full output of the model is available in S1 Table.

**Mean amplitude.** Fig 4 shows the mean amplitude of the target syllables as a function of emotion and Mandarin tone. The main effects of Mandarin tone, $\chi^2(3, N = 8) = 121.1$, $p < .001$, and emotion, $\chi^2(4, N = 8) = 1801.44$, $p < .001$ were observed, but not their interaction $\chi^2(12, N = 8) = 3.92$, $p = .98$. Post hoc pairwise comparisons showed that ANGRY has the highest mean amplitude and NEUTRAL has the lowest. No difference was observed across SAD, HAPPY, and FEAR. Full output of the model is available in S1 Table.

**Duration.** Fig 5 shows the duration of the target syllables as a function of emotion and Mandarin tone. Similar to the mean amplitude, the main effects of Mandarin tone, $\chi^2(3, N = 8) = 32.5$, $p < .001$, and emotion, $\chi^2(4, N = 8) = 639.74$, $p < .001$ were significant but their interaction $\chi^2(12, N = 8) = 14.2$, $p = .29$ was not. Post hoc pairwise comparisons indicate SAD has the longest duration than all other emotions. Full output of the model is available in S1 Table.

## Summary and discussion

Emotions leave a mark on the acoustic characteristics of Mandarin tones. Consistent with previous studies [2, 5–7, 41–53], findings from our acoustic analyses support the observation that emotions shape the acoustic characteristics of speech for both tonal and non-tonal languages [63–65, 97]. Our inclusion of all four Mandarin tones produced by talkers of both sexes further

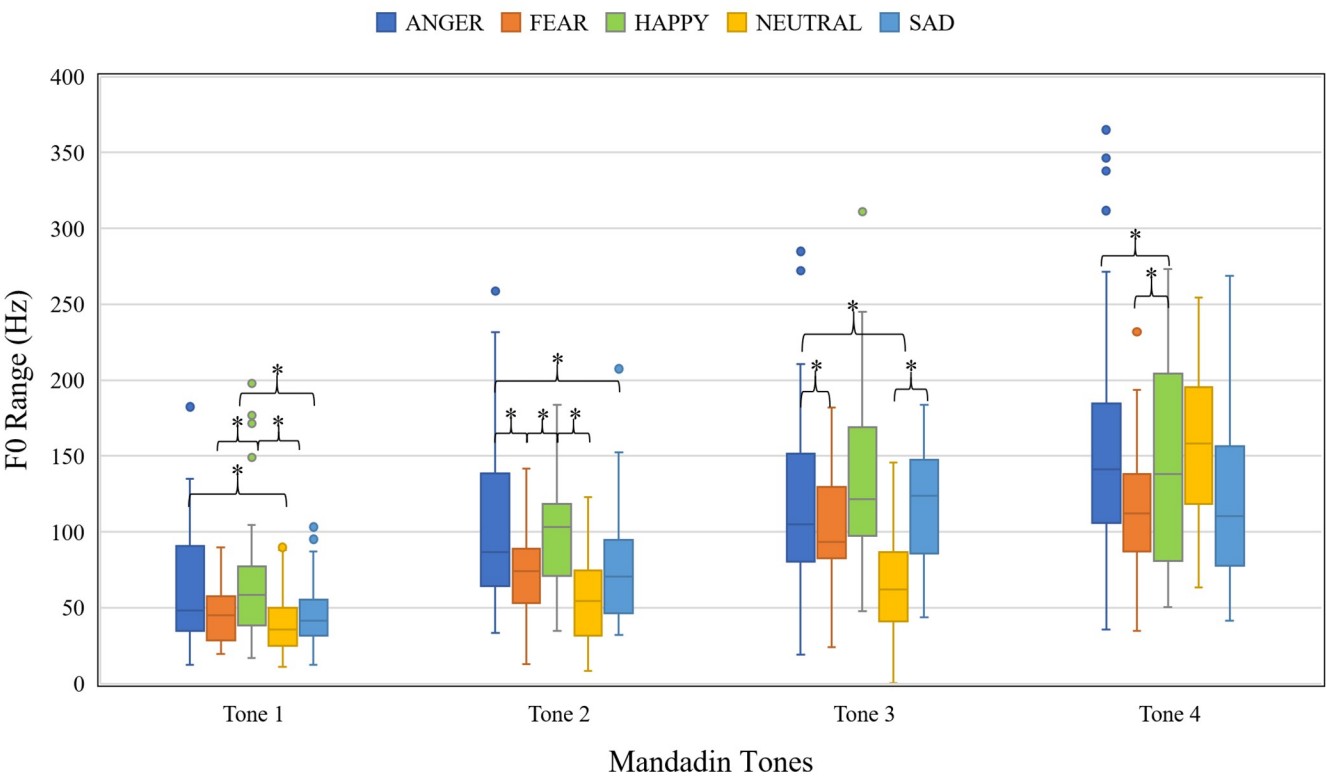

**Fig 3. Boxplot showing the F0 range of target syllables as a function of Mandarin tone and emotion.** ($^*p < .05$).

revealed that the impact of emotions varies depending on specific Mandarin tones. ANGRY has the highest mean F0 and mean amplitude. SAD has the longest duration. In contrast, we did not observe a systematic difference in F0 range.

Table 2 summarizes our findings compared to previous research. There are similarities but also discrepancies. Methodological differences such as talkers (amateurs in previous studies vs. professional actors in the current study) and materials (sentences in previous studies vs. syllables extracted from a carrier phrase in the current study) are likely to have contributed to the discrepancies. We used professional actors in the current study because they are typically more proficient in producing the desired emotions [7, 63]. The presence of the intended emotions was verified in the current study with an independent emotion judgment task. As for materials, since the syllable is the tone-bearing unit in Mandarin, our choice to analyze syllables instead of sentences allowed us to examine the effect of emotions on specific Mandarin tones systematically.

Among the acoustic measures, mean F0 and F0 range appear to yield the most consistent results between previous research and the current study. ANGRY, FEAR, and HAPPY consistently resulted in a higher F0. However, the utility of this measure is difficult to evaluate because of the lack of specificity in the predictions. For example, previous research showed that FEAR and SAD could result in a wider, comparable, or narrower F0 range in Mandarin tones compared to the neutral emotion. Although the current study showed the same results, no consistent patterns could be extracted without taking into consideration specific Mandarin tones.

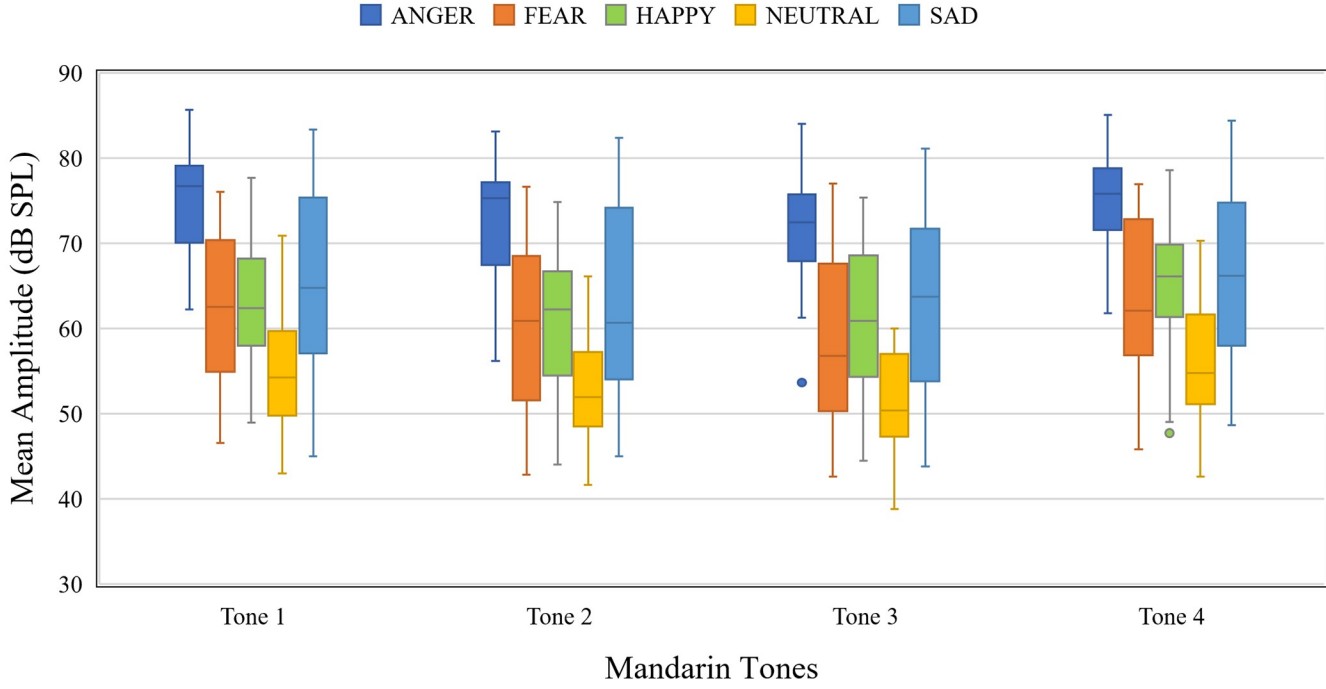

**Fig 4. Boxplot showing the mean amplitude of target syllables as a function of Mandarin tone and emotion.**

Among the emotions, ANGRY consistently results in a higher mean F0, a greater F0 range, and a higher mean amplitude. This finding appears to support the proposal that negative emotions are conveyed more effectively in vocal communication to ensure human survival [43, 78, 79]. If an emotion results in relatively stable acoustic changes in lexical tones, recognition of that emotion is likely to be more robust. However, the other two negative emotions FEAR and SAD did not result in a similar pattern of consistent acoustic changes. It has yet to be determined if ANGRY has a special status among the negative emotions.

## Experiment 2

Findings from Experiment 1 show that emotions shape the acoustic characteristics of Mandarin tones. The extent of the effect also depends on specific Mandarin tones. In Experiment 2, we ask how the acoustic changes induced by emotions would affect the perception of Mandarin tones. Liang and Chen (2019) [71] found that ANGRY resulted in lower accuracy of Mandarin tone identification relative to the neutral emotion. This difference was driven by more accurate identification of Tone 4 compared to Tone 1. We expect to find a similar interaction between Mandarin tones and emotions.

More generally, we evaluate two possibilities regarding how listeners interpret the acoustic signal in terms of Mandarin tones and emotions. On the one hand, Mandarin tone identification may be compromised by emotions because emotions make tones more variable acoustically, and thus more challenging to identify. In this scenario, greater acoustic changes (e.g., those associated with ANGRY) should lead to a less accurate tone identification. On the other hand, Mandarin tone identification may not be compromised by emotions if listeners are able to attribute the acoustic changes into the Mandarin tones and emotions, respectively.

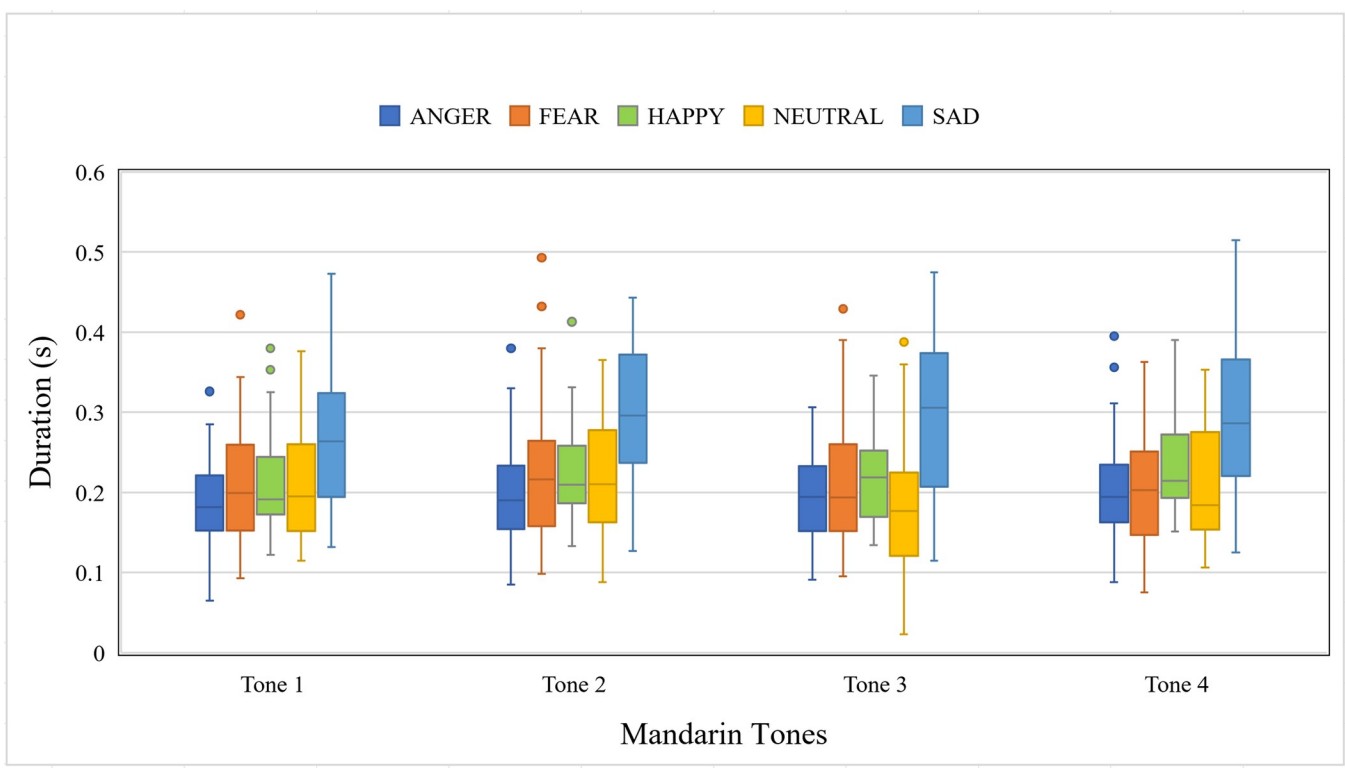

**Fig 5. Boxplot showing the duration of the target syllables as a function of Mandarin tone and emotion.**

Assuming more predictable acoustic changes facilitate identification, more consistent acoustic changes (e.g., those associated with ANGRY) should lead to more accurate tone identification. It should be noted that neither scenarios assume that lexical tones are defined by absolute pitch, amplitude, or duration. Rather, they are two possible ways listeners parse the acoustic signal into distinct sources of acoustic variability.

In addition to Mandarin tone identification, we examine how well the emotions themselves could be recognized from the stimuli. Previous research has shown that negative emotions tend to be identified with higher accuracy than positive emotions. Furthermore, specific Mandarin tones do not seem to affect emotional tone recognition disproportionately [41, 47]. The inclusion of both positive and negative emotions and all four Mandarin tones in the current study would allow us to evaluate these observations.

## Method

**Participants.** This study was reviewed and approved by the Institutional Review Board of National Yang Ming Chiao Tung University (IRB No. 1000063). All participants signed a

**Table 2. Summary of findings from Experiment 1 regarding the effect of emotion relative to the neutral emotion.**

|  | Mean F0 | | F0 range | | Mean amplitude | | Duration | |
|---|---|---|---|---|---|---|---|---|
|  | **Predicted** | **Actual** | **Predicted** | **Actual** | **Predicted** | **Actual** | **Predicted** | **Actual** |
| **ANGRY** | > | > | > or = | > or = | > | > | < | = |
| **FEAR** | > | > | > or = or < | > or = or < | > or < | > | < | = |
| **HAPPY** | > | > or = | > or = | > or = | > or = | > | > or < | > or = |
| **SAD** | < or = | > | > or = or < | > or = | < | > | > | > |

written informed consent. No minors participated in this study. No medical records or archived samples were used in this study.

Thirty-six adults (23 women and 13 men) with ages ranging from 19 to 23 years (mean age 20.08 ± 0.91 years) participated in Experiment 2. All participants were native speakers of Taiwan Mandarin and reported no known history of speech and hearing disorders. Each participant was paid $1,000 NTD ($34 USD) for their participation. All participants passed a screening of Mandarin tone identification in the neutral emotion presented in isolation and in context. Identification accuracy of Mandarin Tone 1, Tone 2, Tone 3, and Tone 4 was 97%, 93%, 93%, and 98% in isolation, and 100%, 100%, 99%, and 100% in context.

**Stimuli.** The stimuli used in this experiment were selected from one female and one male talker among the eight talkers who recorded the stimuli for Experiment 1. The talkers who received the highest rating in the emotion judgment task (reported in Experiment 1) in their respective sex group were chosen. For tone identification, all five emotions were included, resulting in 240 stimuli (3 syllables, 4 tones, 5 emotions, 2 repetitions, and 2 talkers). The 240 stimuli were presented in isolation or embedded in the carrier phrase, resulting in a total of 480 trials. For emotion recognition, the NEUTRAL stimuli and response option were excluded to prevent participants from using NEUTRAL as a default response for stimuli that they were uncertain about. As a result, there were 192 stimuli (3 syllables, 4 tones, 4 emotions, 2 repetitions, and 2 talkers). The 192 stimuli were presented in isolation or embedded in the carrier phrase, resulting in a total of 384 trials.

**Procedure.** This experiment took place in a sound-treated booth in the Department of Biomedical Engineering at National Yang Ming Chiao Tung University. The LabVIEW program (National Instruments) on a Windows 10 laptop computer was used for stimulus delivery and response acquisition. Stimuli were presented at each participant's preferred hearing level over a pair of Beyerdynamic DT 990 PRO headphones.

The participant's task was to listen to each stimulus and identify the Mandarin tone and the emotion of the target syllable. The stimuli were presented in four blocks in the following order: (1) isolated syllables for Mandarin tone identification; (2) isolated syllables for emotion recognition; (3) target syllables embedded in the carrier phrase for Mandarin tone identification; and (4) target syllables embedded in the carrier phrase for emotion recognition. The order of stimuli within each block was randomized for each participant. The Random Number Generator in LabVIEW was then used to generate a unique presentation order for each participant. Brief breaks were provided between the blocks.

For Mandarin tone identification, four response buttons marked with "一聲 (Tone 1)", "二聲 (Tone 2)", "三聲 (Tone 3)", and "四聲 (Tone 4)" were displayed at the four corners of the computer screen and equidistant from the center of the screen. For emotion recognition, four response buttons marked with "生氣 (ANGRY)", "害怕 (FEAR)", "快樂 (HAPPY)", and "傷心 (SAD)" were displayed at the four corners of the computer screen and equidistant from the center of the screen. The NEUTRAL response option was not included to avoid listener using NEUTRAL as a default for stimuli that they were not sure about. At the beginning of each trial, a cursor appeared briefly at the center of the screen, followed by the auditory stimulus. Listeners responded by clicking one of the four buttons on the screen using a computer mouse. The next trial was then presented 500 ms after the response. If participants were not sure about the tone identity, they were told to make their best guess. Each experimental session took approximately 60–90 minutes to complete.

## Results

**Mandarin tone identification.** Fig 6 shows the accuracy of Mandarin tone identification as a function of emotion, Mandarin tone, and context. Overall, Mandarin tones were identified

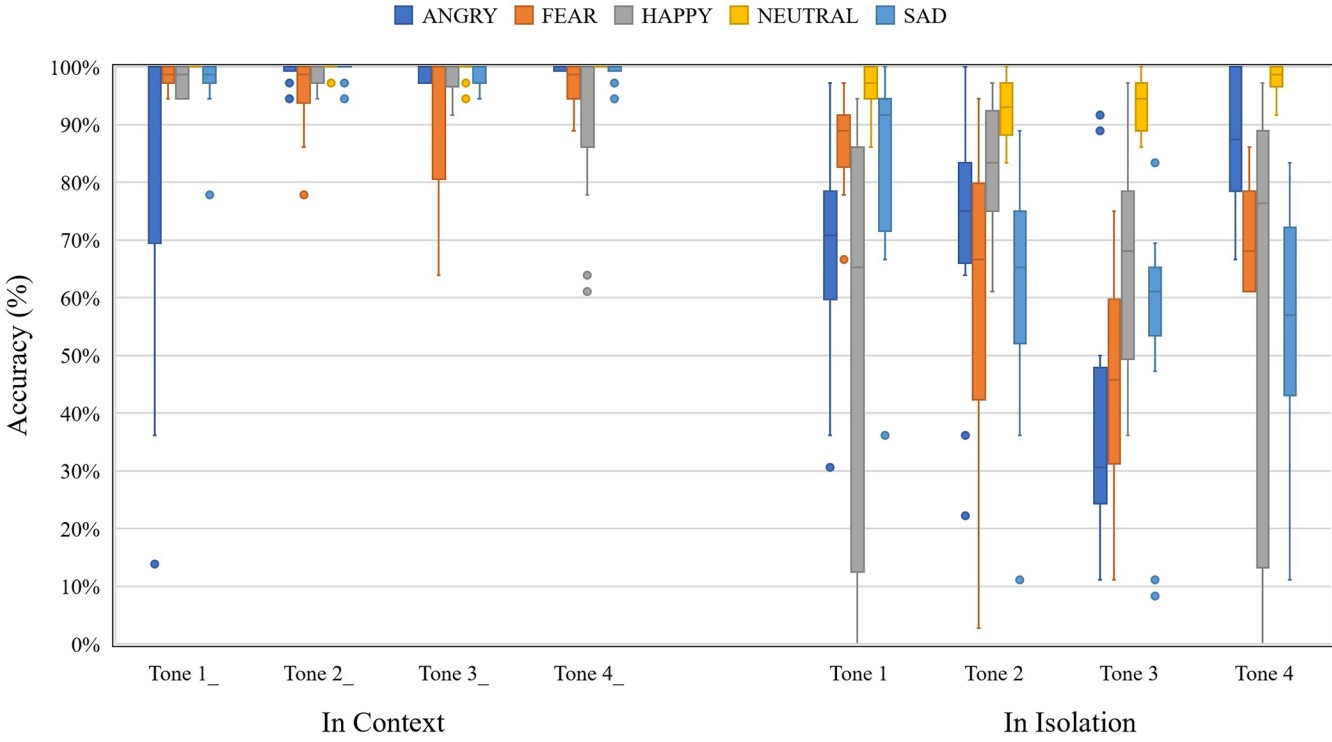

**Fig 6. Boxplot showing the Mandarin tone identification accuracy as a function of Mandarin tone, emotion, and context.**

more accurately in context, and the effect of emotion appeared to be much more variable when the target syllable was presented in isolation. For example, all four Mandarin tones were identified quite well in NEUTRAL regardless of context, but the presence of other emotions compromised the identification of isolated Mandarin tones disproportionately compared to those presented in context.

To evaluate these observations statistically, a mixed-effect logistic regression model was fitted to the Mandarin tone identification data. Mandarin tone (T1, T2, T3, and T4), emotion (ANGRY, FEAR, HAPPY, SAD, and NEUTRAL), context (in isolation and in context), and the tone-emotion interaction were entered into the model as fixed effects. Talker sex, syllable type, and repetition were entered as random effect. The dependent variable was the binary Mandarin tone identification (i.e., correct and incorrect). Full output of the model is available in S2 Table.

All main effects were significant: Mandarin tone, $\chi^2(3, N = 36) = 83.12, p < .001$; emotion, $\chi^2(4, N = 36) = 486.38, p < .001$; and context, $\chi^2(1, N = 36) = 1534.71, p < .001$. The tone-emotion interaction was also significant: Mandarin tone-emotion, $\chi^2(12, N = 36) = 492.89, p < .001$. Post hoc pairwise comparisons indicate that tone identification accuracy varied across different emotions (Table 3).

To examine the specific types of Mandarin tone identification errors, Tables 4 and 5 show confusion matrices of Mandarin tone identification responses for syllables presented in isolation (Table 4) and in context (Table 5). To facilitate interpretation of the confusion patterns, the most common error for each Mandarin tone that exceeds 20% is highlighted in gray. For syllables presented in isolation (Table 4), Mandarin tones produced with NEUTRAL were rarely misidentified as other tones. There appears to be a response bias where Mandarin tones,

**Table 3. Summary of pairwise comparisons for Mandarin tone identification accuracy as a function of emotion.** The symbol > indicates a significant difference ($p < .05$) and = indicates no significant difference.

| Emotion | Mandarin tone |
|---------|---------------|
| **ANGRY** | T4 > T2 > T1 = T3 |
| **FEAR** | T1 > T4 > T2 > T3 |
| **HAPPY** | T2 > T3 > T4 = T1 |
| **NEUTRAL** | T4 = T1, T4 > T3, T4 > T2, T1 > T2 |
| **SAD** | T1 > T2 = T3 = T4 |

particularly Tones 1 and 3, were most misidentified as Tone 4 when presented in an ANGRY tone of voice. Similarly, there appears to be a response bias for tones to be identified as Tone 1 when presented in a FEAR tone of voice. In contrast, the errors for Mandarin tones produced with HAPPY or SAD did not show distinct bias patterns.

For syllables presented in context (Table 5), Mandarin tone identification was remarkably accurate. There were only three instances where the accuracy fell below 90%, and only one of those was below 80%. In terms of confusion patterns, there was only one error that approached 20%, where an ANGRY Tone 1 was misidentified as Tone 4. There were no other dominant errors. In sum, the presence of the carrier phrase effectively neutralized the negative impact of emotions on Mandarin tones identification from isolated syllables.

## Emotion recognition

Fig 7 shows the accuracy of emotion recognition as a function of emotion, Mandarin tone, and context. Overall, accuracy appears higher for target syllables presented in context than in

**Table 4. Confusion matrices of Mandarin tone identification responses for syllables presented in isolation across emotions.** The most common error that exceeds 20% for each emotion is highlighted in gray.

| Emotion | Mandarin tone stimulus | Mandarin tone response | | | |
|---------|------------------------|------------------------|---------|---------|---------|
| | | **Tone 1** | **Tone 2** | **Tone 3** | **Tone 4** |
| **Neutral** | **Tone 1** | **97%** | 2% | 0% | 1% |
| | **Tone 2** | 0% | **93%** | 7% | 0% |
| | **Tone 3** | 2% | 2% | **93%** | 3% |
| | **Tone 4** | 1% | 1% | 0% | **98%** |
| **Angry** | **Tone 1** | **68%** | 12% | 2% | 18% |
| | **Tone 2** | 11% | **71%** | 16% | 2% |
| | **Tone 3** | 18% | 6% | **40%** | 36% |
| | **Tone 4** | 9% | 1% | 3% | **87%** |
| **Fear** | **Tone 1** | **86%** | 10% | 2% | 2% |
| | **Tone 2** | 26% | **58%** | 15% | 1% |
| | **Tone 3** | 25% | 13% | **46%** | 16% |
| | **Tone 4** | 20% | 2% | 8% | **70%** |
| **Happy** | **Tone 1** | **52%** | 41% | 3% | 4% |
| | **Tone 2** | 5% | **83%** | 11% | 1% |
| | **Tone 3** | 4% | 5% | **66%** | 25% |
| | **Tone 4** | 32% | 7% | 5% | **56%** |
| **Sad** | **Tone 1** | **83%** | 13% | 3% | 1% |
| | **Tone 2** | 13% | **61%** | 25% | 1% |
| | **Tone 3** | 13% | 13% | **60%** | 14% |
| | **Tone 4** | 24% | 4% | 17% | **55%** |

**Table 5. Confusion matrices of Mandarin tone identification responses for syllables presented in context across emotions.**

| Emotion | Mandarin tone stimulus | Mandarin tone response | | | |
|---|---|---|---|---|---|
| | | Tone 1 | Tone 2 | Tone 3 | Tone 4 |
| Neutral | Tone 1 | 100% | 0% | 0% | 0% |
| | Tone 2 | 0% | 100% | 0% | 0% |
| | Tone 3 | 0% | 1% | 99% | 0% |
| | Tone 4 | 0% | 0% | 0% | 100% |
| Angry | Tone 1 | 78% | 1% | 2% | 19% |
| | Tone 2 | 0% | 98% | 2% | 0% |
| | Tone 3 | 0% | 1% | 98% | 1% |
| | Tone 4 | 3% | 0% | 0% | 97% |
| Fear | Tone 1 | 96% | 2% | 1% | 1% |
| | Tone 2 | 4% | 93% | 2% | 1% |
| | Tone 3 | 3% | 8% | 88% | 1% |
| | Tone 4 | 7% | 1% | 1% | 91% |
| Happy | Tone 1 | 92% | 1% | 1% | 6% |
| | Tone 2 | 2% | 96% | 1% | 1% |
| | Tone 3 | 1% | 2% | 97% | 1% |
| | Tone 4 | 11% | 2% | 0% | 87% |
| Sad | Tone 1 | 95% | 3% | 0% | 2% |
| | Tone 2 | 1% | 97% | 2% | 0% |
| | Tone 3 | 1% | 4% | 95% | 0% |
| | Tone 4 | 1% | 2% | 1% | 96% |

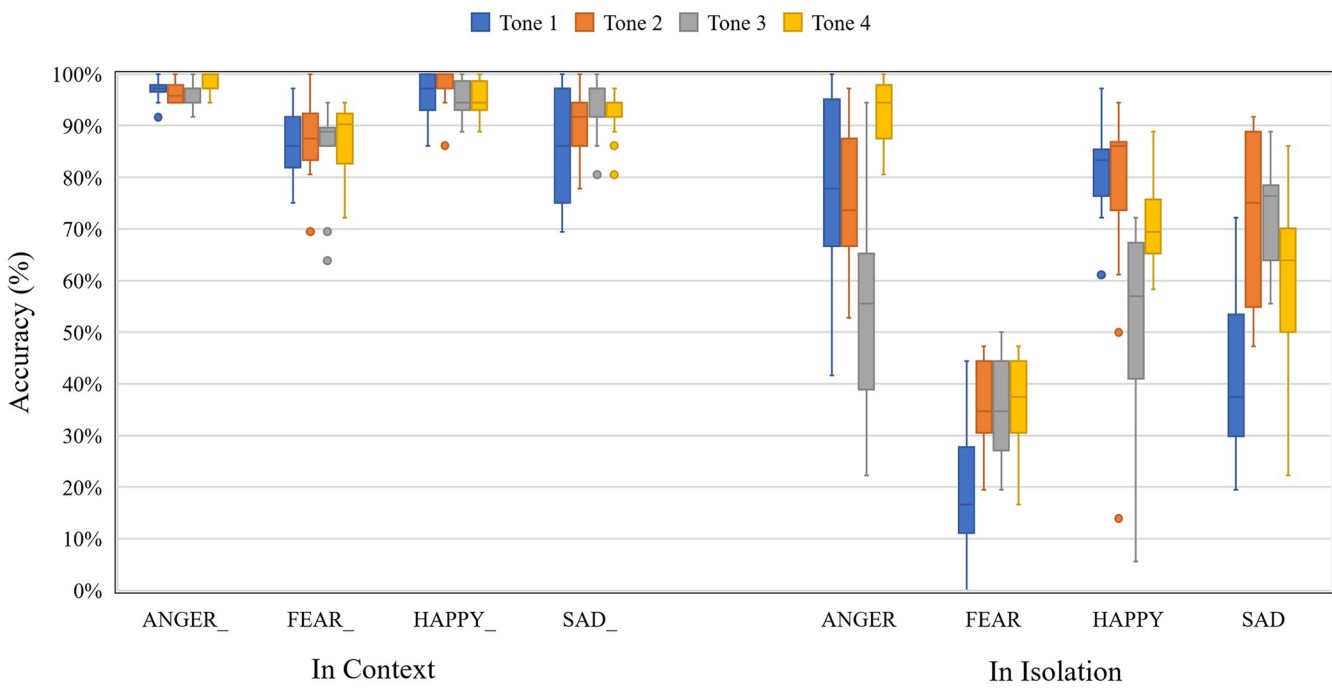

**Fig 7. Boxplot showing the emotion recognition accuracy as a function of Mandarin tone and context.**

**Table 6. Summary of pairwise comparisons for emotion recognition accuracy as a function of Mandarin tone.** The symbol > indicates a significant difference ($p <$ .05) and = indicates no significant difference.

| Mandarin tone | Emotion |
|---|---|
| Tone 1 | ANGRY = HAPPY > SAD > FEAR |
| Tone 2 | ANGRY = HAPPY > SAD > FEAR |
| Tone 3 | SAD > ANGRY = HAPPY > FEAR |
| Tone 4 | ANGRY > HAPPY > SAD > FEAR |

isolation. Importantly, the four emotions appear to be affected by context to different degrees. For example, FEAR appears to be identified disproportionately worse in isolation.

A mixed-effect logistic regression model with Mandarin tone (T1, T2, T3, and T4), emotion (ANGRY, FEAR, HAPPY, and SAD), context (in isolation and in context), and the tone-emotion interaction as fixed effects, talker sex, syllable type, and repetition as random effects were constructed. The dependent variable was the binary emotion recognition (correct and incorrect). Full output of the model is available in S3 Table.

All main effects were significant: Mandarin tone, $\chi^2(3, N = 36) = 92.92$, $p <$ .001; emotion, $\chi^2(1, N = 36) = 775.08$, $p <$ .001; and context, $\chi^2(1, N = 36) = 1843.42$, $p <$ .001. The tone-emotion interaction was also significant: $\chi^2(9, N = 36) = 327.95$, $p <$ .001. Post hoc pairwise comparisons indicate that emotion recognition accuracy varied across different Mandarin tones (Table 6). For example, ANGRY was recognized most accurately in all but Mandarin Tone 3, whereas FEAR was recognized least accurately in all Mandarin tones.

To examine the types of emotion recognition errors made by listeners, Tables 7 and 8 show confusion matrices of emotion recognition responses for syllables presented in isolation (Table 7) and in context (Table 8). To facilitate the interpretation of the confusion patterns, the most common error for each emotion that exceeds 20% is highlighted in gray .

For isolated syllables (Table 7), emotion recognition accuracy ranged from 21% to 93%. There appears to be a response bias where SAD stimuli were most commonly identified as FEAR in Tone 1 and 4. Similarly, there appears to be a response bias where FEAR stimuli were commonly identified as HAPPY in Tones 1 and 2, but as SAD in Tones 3 and 4. In contrast, ANGRY and HAPPY stimuli did not result in any dominant bias patterns.

For syllables presented in context (Table 8), emotions were recognized more accurately, ranging from 85% to 99%. In terms of confusion patterns, the only error that exceeded 10% was the SAD response to the FEAR stimulus in Tone 1 (15%). Because of the high accuracy, none of the emotions resulted in any dominant bias patterns.

## Summary and discussion

As predicted, Mandarin tone identification accuracy varied across emotions, i.e., not all tones were identified equally well for different emotions. We replicated Liang and Chen's (2019) [71] finding that Tone 4 is identified more accurately than Tone 1 in ANGER. Our results further revealed additional contingencies on emotion (Table 3). Tone identification accuracy also varied across contexts. All tones were identified more accurately when the target syllables were embedded in the carrier phrase, but not all tones were identified equally well across the two contexts (Table 3). For syllables presented in isolation, confusion analyses showed that Tone 4 was the most common error for the ANGRY stimuli, and Tone 1 was the most common error for the FEAR stimuli (Table 4). For syllables presented in the carrier phrase, confusion analyses did not reveal any notable patterns except that Tone 4 was a common error for ANGRY Tone 1 stimuli (Table 5).

**Table 7. Confusion matrices of emotion recognition responses for syllables presented in isolation.** The most common error that exceeds 20% for each emotion is highlighted in gray.

| Mandarin tone | Emotional tone stimulus | Emotion response | | | |
|---|---|---|---|---|---|
| | | ANGRY | FEAR | HAPPY | SAD |
| Tone 1 | ANGRY | 77% | 9% | 13% | 1% |
| | FEAR | 11% | 21% | 56% | 12% |
| | HAPPY | 8% | 6% | 82% | 4% |
| | SAD | 4% | 29% | 25% | 42% |
| Tone 2 | ANGRY | 76% | 5% | 17% | 2% |
| | FEAR | 14% | 36% | 29% | 21% |
| | HAPPY | 5% | 8% | 76% | 11% |
| | SAD | 2% | 21% | 5% | 72% |
| Tone 3 | ANGRY | 53% | 15% | 9% | 23% |
| | FEAR | 7% | 34% | 13% | 46% |
| | HAPPY | 12% | 12% | 49% | 27% |
| | SAD | 2% | 22% | 3% | 73% |
| Tone 4 | ANGRY | 93% | 1% | 3% | 3% |
| | FEAR | 11% | 37% | 24% | 28% |
| | HAPPY | 8% | 11% | 71% | 10% |
| | SAD | 6% | 29% | 4% | 61% |

The hypothesis that ANGRY stimuli would result in less or more accurate tone identification depending on how listeners interpret the acoustic signal was not supported by our data. Although ANGRY resulted in the greatest acoustic changes (as shown in Experiment 1), Mandarin tones produced with ANGRY were not identified disproportionately worse. Similarly, although ANGRY was associated with the most consistent acoustic changes (as shown in Experiment 1), Mandarin tones produced with ANGRY were not identified disproportionately

**Table 8. Confusion matrices of emotion recognition responses for syllables presented in context.**

| Mandarin tone | Emotional tone stimulus | Emotional tone response | | | |
|---|---|---|---|---|---|
| | | ANGRY | FEAR | HAPPY | SAD |
| Tone 1 | ANGRY | 97% | 1% | 1% | 1% |
| | FEAR | 1% | 87% | 6% | 6% |
| | HAPPY | 0% | 2% | 96% | 2% |
| | SAD | 0% | 15% | 0% | 85% |
| Tone 2 | ANGRY | 97% | 2% | 0% | 1% |
| | FEAR | 1% | 88% | 4% | 7% |
| | HAPPY | 0% | 1% | 97% | 2% |
| | SAD | 0% | 8% | 1% | 91% |
| Tone 3 | ANGRY | 96% | 2% | 0% | 1% |
| | FEAR | 1% | 85% | 7% | 7% |
| | HAPPY | 1% | 2% | 96% | 1% |
| | SAD | 0% | 6% | 0% | 94% |
| Tone 4 | ANGRY | 99% | 1% | 0% | 0% |
| | FEAR | 1% | 87% | 3% | 9% |
| | HAPPY | 1% | 2% | 95% | 2% |
| | SAD | 1% | 7% | 0% | 92% |

better. With the extensive interactions between tone, emotion, and context, the current data is inconclusive as to how listeners parsed the acoustic signal into distinct sources.

Regarding emotion recognition, accuracy of emotion recognition varied across Mandarin tones, i.e., not all emotions were recognized equally well in different Mandarin tones (Table 6). However, ANGRY was recognized most accurately in three of the four Mandarin tones. Like Mandarin tones, emotions were recognized more accurately when the target syllables were embedded in the carrier phrase. Unlike Mandarin tones, the order of emotion identification accuracy was more consistent between in isolation and in context (Table 6), i.e., ANGRY was recognized better than HAPPY, followed by SAD, and FEAR was recognized least accurately. For syllables presented in isolation, confusion analyses showed that the most common error for SAD stimulus was FEAR response (Table 7). For syllables presented in context, confusion analyses did not reveal any notable error patterns (Table 8).

Taken together, data from Mandarin tone identification and emotion recognition revealed several similarities. Mandarin tones identification accuracy varied across emotions, just as emotions recognition accuracy varied across Mandarin tones. In addition, Mandarin tone identification and emotion recognition were both more accurate with syllables presented in the carrier phrase. The presence of the carrier phrase facilitated both Mandarin tone identification and emotion recognition, especially for those tones and emotions that were identified poorly in isolated syllables. Finally, for both Mandarin tone identification and emotion recognition, there were distinct error patterns in isolated syllables, but barely any notable error patterns in context.

There are also differences between Mandarin tone identification and emotion recognition. First, the accuracy of identifying specific Mandarin tones varied greatly across different emotions (Table 3), but the accuracy of recognizing specific emotions was more consistent across different Mandarin tones. For example, ANGRY was consistently identified better than most other emotions, and FEAR was consistently identified worse than all other emotions (Table 6). In other words, how well a Mandarin tone is identified depends heavily on specific emotions, but how well an emotion is identified does not depend as much on specific Mandarin tones. At first sight, this finding appears to suggest that emotion recognition is more robust than Mandarin tone identification. However, the accuracy of emotion recognition (ranging from 21% to 93%) was not higher than Mandarin tone identification (40% to 98%) in isolated syllables. The accuracy in the carrier phrase also seemed comparable: emotion recognition (85% to 99%); Mandarin tone identification (78% to 100%). Emotions do not seem to be inherently easier to identify compared to Mandarin tones. Rather, our data suggest that their mutual influence is asymmetrical.

Furthermore, Mandarin tone identification and emotion recognition differed in their interaction with context. The accuracy of identifying specific Mandarin tones varied substantially between the two contexts (Table 3), but the accuracy of recognizing specific emotions was quite consistent between the two contexts (Table 6).

## Conclusions, limitations, and future directions

Three conclusions can be drawn from our findings. First, acoustic characteristics of Mandarin tones are shaped by emotions in complex but systematic ways, depending on specific Mandarin tones and specific emotions. Second, emotions affect Mandarin tone identification to a greater extent than Mandarin tones affect emotion recognition. Finally, the presence of carrier phrase facilitates both Mandarin tone identification and emotion recognition.

There are a number of limitations of the study. First, this study was conceptualized with four-emotion models; therefore the experimental materials were recorded with only four basic

emotions. As noted in the introduction, there are different opinions as to the number of basic emotions, and basic-emotion models do not fully capture valence and arousal in the dimensional approach to emotion. For example, of the four emotions used in this study, only happiness is a positive emotion, and only sadness is considered low in arousal. Inclusion of a wider range of emotions would have allowed a more nuanced examination of the interaction between lexical and emotional tones. Second, only one carrier phrase was used; i.e., we did not systematically manipulate the tone that precedes the target syllable like Liang and Chen (2019) [71] did. It is not known whether our results would generalize to other tonal contexts. Third, the stimuli were produced by speakers of Taiwan Mandarin, and the participants in the perception experiment were also Taiwan Mandarin speakers. It is not known whether our results would generalize to other variants of Mandarin. Fourth, we did not include NEUTRAL emotion in the stimuli for the emotion recognition task. Since NEUTRAL emotion was included in the acoustic analysis, it would have been informative to include it in the perception experiment. Fifth, the syllables presented in isolation in the current study were excised from a carrier phrase; therefore, it is not known whether the findings would generalize to syllables produced in isolation. Six, since syllables in isolation were presented before syllables in context, familiarization from the isolation could have boosted response accuracy in the context blocks. Finally, we used the neutral condition as a baseline for emotion-related comparisons. Comparisons among the other emotions could generate further insights.

In addition to addressing these limitations, future studies could consider the following extensions. First, although the acoustic measures used in this study were commonly used in research on lexical tones and emotions, additional acoustic measures on voice quality and quantitative measures of F0 contour (e.g., Functional Data Analysis [94, 95]) would be useful. Second, although our use of naturally produced stimuli in the perception experiment preserved all the acoustic cues available to listeners, it was not possible to isolate the contributions of specific acoustic cues. Future studies could systematically manipulate those cues to identify their individual contributions to lexical tone identification and emotion recognition. Third, although our data revealed *how* the acoustics and perception of lexical tones depend on factors including emotions and context, it is not clear *why* these factors affect lexical tone perception in such complex ways. Finally, the asymmetry in the mutual influence between lexical tones and emotions highlights the need for further research on how tonal language experience shapes lexical tone identification [71] and emotion recognition.

## Supporting information

**S1 Table. Full output of a linear mixed-effect logistic regression model for acoustic analysis of four Mandarin tones with five emotions.**
(DOCX)

**S2 Table. Full output of a linear mixed-effect logistic regression model for Mandarin tone identification.**
(DOCX)

**S3 Table. Full output of a linear mixed-effect logistic regression model for emotion recognition.**
(DOCX)

**S4 Table. All data of two experiments.**
(RTF)

## Acknowledgments

We thank Dr. Liquan Liu, Dr. Chong Chee Seng, Dr. Martijn Goudbeek, and anonymous reviewers for their constructive feedback. We also thank Faith Fedele for editorial assistance.

## Author Contributions

**Data curation:** Xianhui Wang.

**Investigation:** Hui-Shan Chang, Shuenn-Tsong Young, Woei-Chyn Chu.

**Methodology:** Hui-Shan Chang, Shuenn-Tsong Young, Woei-Chyn Chu.

**Resources:** Cheng-Hsuan Li.

**Software:** Cheng-Hsuan Li.

**Writing – original draft:** Hui-Shan Chang.

**Writing – review & editing:** Chao-Yang Lee, Shuenn-Tsong Young, Woei-Chyn Chu.

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
