## [Decision Letter · Decision Letter 0]

24 Nov 2021

PONE-D-21-23777Emotional tones of voice affect the acoustics and perception of Mandarin tonesPLOS ONE

Dear Dr. Chu,

Thank you for submitting your manuscript to PLOS ONE. After careful consideration, we feel that it has merit but does not meet PLOS ONE’s publication criteria as it currently stands. Therefore, we invite you to submit a revised version of the manuscript that addresses the points raised during the review process.

Specifically, the reviewers acknowledge the merits of this study but see major problems with the integration with the current literature as well as with the justification of the design and the appropriateness of the analyses. Because PLOS ONE emphasises methodological soundness over novelty and impact I am minded to give you the opportunity to address the reviewers' comments very carefully. Reviewer 2 makes particularly pertinent and detailed comments that I encourage you to address. I particularly see the need to address the comments regarding ANOVA as an unsuitable statistical approach to these data. In addition, please make sure the data are being made available to the reviewers - you state that they are but the reviewers had trouble finding them. Finally, I would like to point out that request of revision does not constitute guarantee of acceptance of the revised version.

We look forward to receiving your revised manuscript.

Kind regards,

Vera Kempe

Academic Editor

PLOS ONE

Journal Requirements:

2. Please change "female” or "male" to "woman” or "man" as appropriate, when used as a noun (see for instance https://apastyle.apa.org/style-grammar-guidelines/bias-free-language/gender).

4. We note that you have referenced (ChangLiao I [31]) which has currently not yet been accepted for publication. Please remove this from your References and amend this to state in the body of your manuscript: (ChangLiao I . [Unpublished]”) as detailed online in our guide for authors http://journals.plos.org/plosone/s/submission-guidelines#loc-reference-style

Reviewers' comments:

Reviewer's Responses to Questions

**Comments to the Author**

1. Is the manuscript technically sound, and do the data support the conclusions?

Reviewer #1: Partly

Reviewer #2: Partly

2. Has the statistical analysis been performed appropriately and rigorously? 

Reviewer #1: Yes

Reviewer #2: No

3. Have the authors made all data underlying the findings in their manuscript fully available?

Reviewer #1: No

Reviewer #2: Yes

4. Is the manuscript presented in an intelligible fashion and written in standard English?

Reviewer #1: Yes

Reviewer #2: No

5. Review Comments to the Author

Reviewer #1: The study is interesting and relevant. I enjoy reading it very much.

The separation between acoustics and perception is wonderful.

Please find the points of improvement/discussion below.

Overall review questions:

1. Is the manuscript technically sound, and do the data support the conclusions?

Narratives can appear more concisely throughout the manuscript.

If possible, a bit more discussion on the other way (lexical affecting emotional tone) would be appreciated.

There is a potential design issue in the “context” condition, which can be resolved by careful discussion since the experiments have been done.

Based on the previous comment, please double-check the research questions. Make sure the questions asked are the variables being examined.

2. Has the statistical analysis been performed appropriately and rigorously?

The effect size is omitted in the report.

3. Have the authors made all data underlying the findings in their manuscript fully available?

Maybe I missed this: it is unclear where (anonymous) data are stored.

4. Is the manuscript presented in an intelligible fashion and written in standard English?

The manuscript is understandable but is encouraged to be double-checked on its English writing style. There is space for further improvement.

Detailed comments:

Abstract

1. Polish writing, including style, words and grammar

A. The starting sentence can be polished to enhance the significance of the study

B. “the effect of emotional tones on Mandarin tone acoustics varied depending on specific Mandarin tones, specific emotional tones, and talker gender.” Please rephrase.

C. “select materials”: selected? (same in Ln283)

D. “Emotional tones also affected the four acoustic measures to different degrees.” Please integrate with the narratives prior.

2. Reflecting on the narratives of Exp.2, clarify whether Exp.1 is presented in isolation or in context.

Introduction

Ln48

I understand that the authors adopted a four-basic emotion model. I wonder why the more commonly accepted six-basic emotion model, adding surprise and disgust in the study, is not considered.

Ln51

Please double check "amplitude" or "intensity"

Ln70

1. I have the feeling that these scenarios are restricted to certain context/isolation, or when only key cues are involved. This is later discussed in Ln575.

2. I wonder if these are more like logical possibilities unless the authors argue that different scenarios may occur with different emotional-lexical tone combos.

Ln85

I wonder whether the other way around is worth highlighting, namely, how lexical tone affects emotion (e.g., emotional tone) perception? This is analysed and discussed after all (e.g., Ln805, later asymmetry)

Ln192

Does falling tone affect anger perception?

Ln198-212

Some information here seems to have been mentioned earlier.

Ln232, 234

Wang et al. - repetitive presentation

Ln261

It is unclear why lexical tones lacking F0 variation would be predicted to make the emotional tone less perceptible in the first place. Can it be the other way around? That is, since lexical tone has no variation, would variations of emotion prevail?

Experiment/Analyses

Ln310

Some tables are a little bit difficult to process although the meaning is clear. I do not have suggestions here because the authors have already given careful thoughts on the presentation of these tables.

Ln318

Why is 4 a good number of participants as the baseline for the acoustic characteristics of emotional tone?

Ln537-540

Please rephrase to reduce repetitiveness.

Ln566

Please rephrase "In addition, we also"

Discussion

Regarding "Context":

There can be several possibilities to understand this word in the “context” of the current paper. It may be worthwhile specifying which context the authors are referring to (carrier sentences).

Also, what the context adds to/change the picture needs to be spelt out more overtly (e.g., ease of semantic parsing and/or tonal coarticulation? If one, why not the other?)

One methodological issue is to what extent the discrepancy between context emotion (no emotion if I understand correctly) and target word emotion would affect the current findings in the context condition. This needs to be discussed.

Reviewer #2: 1. Is the manuscript technically sound, and do the data support the conclusions?

a. The current study vs. directly relevant past studies: This study is a potentially very valuable addition to the line of research on the influence of emotion on the production and perception of lexical tones in Mandarin. Different from past studies, the present study looks at the influence of emotion on lexical tones in both production and perception. However, this does not mean that past work with a focus on either the production side of the coin or the perception should be disconnected from the current study. In other words, the authors did not seem to make a sufficient use of existent findings to motivate their study and develop the hypotheses and predictions. Specially, the authors focused on how the realisation of each of the five basic emotions impacts the f0 mean, f0 range, intensity and duration of the target syllable. The literature review 'how emotional tones affect speech acoustics' only discussed the differences in the degree to which an acoustic parameter is exploited to express emotions between tone and non-tone languages. The authors referred to 'the literature reviewed' when presenting their predictions for changes in f0, duration and intensity (lines 300-302). But past studies on how emotion expression influences the f0 and duration of each lexical tone in Mandarin was not reviewed. Furthermore, although research on the effect of emotion expression on the perception of Mandarin tones by native speakers of Mandarin and second language learners of Mandarin is small in numbers, there has been some published work on this topic, e.g. Liang & Chen (2019), which was published in the ICPhS 2019 proceedings. This work was mistakenly cited as Liang and Chen's MA thesis in the references section, summarised in the section 'How emotional tones affect speech perception' and subsequently dismissed. This is a pity because the authors could have built their work on the findings from Liang and Chen, especially considering the fact that they had the same research question on the production side as Liang and Chen, and used a similar method in both Experiment 1 and Experiment 2 to Liang and Chen’s methodology. The main differences concerning production elicitation seems to be that Liang and Chen used phototactically legal non-words as target words and systematically varied the tonal contexts preceding the monosyllabic target words whereas the current study used real words and had only one tonal context, i.e. the monosyllabic target words were preceded by tone 1 in the carried sentence. Liang and Chen's findings can thus provide a very useful starting point for the current study in terms of both hypothesis formulation and methodology.

b. Taiwan Mandarin is prosodically not the same as Beijing Mandarin (e.g. in prosodic realisation of focus), even though they both have the same four lexical tones. This difference is not sufficiently acknowledged throughout the study. Notably, the similarities in findings between this study and those from Liang and Chen (2019) can have interesting implications for the prosodic similarities and differences regarding the interaction between emotion and lexical tones between Taiwan Mandarin and Beijing Mandarin.

c. Experiment 1: The participants were specifically instructed not to place 'excessive stress on the target syllables'. This seems to suggest that the participants have no uniformed way to determine the information structure of each target sentence. They could have treated each target sentence as a response to a 'what-happens'-like question, rendering the whole sentence focal (broad focus) or treated the target word as the focus (narrow focus) in spite of the instructions. What the participants did regarding the information structure of the target sentences may influence how they used prosody to realise emotion and tones. For example, they might have distributed the prosodic changes over the whole sentence if they had a broad-focus analysis, or the prosodic changes might have concentrated in the target syllables if they had a narrow-focus analysis.

d. Experiment 1: The recordings from the participants were rated by 30 native speakers of Taiwan Mandarin. No information was given on who these participants were and how they were recruited. Were they the same participants who did Experiment 2?

e. Experiment 2: Experiment 2 hinged on two opposing assumptions. Neither is well motivated. One of the assumptions was that ‘emotional tones alter the canonical acoustic characteristics of Mandarin tones, making it difficult for listeners to retrieve the Mandarin tones’ (Lines 554-556). The authors did not specify anywhere in the manuscript what the ‘canonical acoustic characteristics of Mandarin tones’ should be. I do not think that past studies have provided evidence for such a claim either. Instead, it has been shown that the changes in the acoustic realisation of lexical tones related to the expression of emotions are restricted to the gradient changes in terms of higher of lower mean pitch, smaller or wider pitch range, or longer or shorter duration. Such changes are phonetic by nature because lexical tones are not defined by absolute pitch, duration and intensity. The other assumption is that ‘listeners are able to disaggregate the acoustic changes into two sources – Mandarin tones and emotional tones’. This assumption is itself based on the assumptions that lexical tones have absolute f0, duration and intensity values, which is not the case, and the perception of emotion prosody is categorical, for which evidence has emerged (e.g. Laukka 2005).

Laukka, P. (2005). Categorical perception of vocal emotion expressions. Emotion, 5(3), 277–295.

f. The factors ‘context’ vs. ‘isolation’ in Experiment 2: The target words were presented to the participants in both their original carrier sentence and out of their original carrier sentence (e.g. cut out of the carrier sentences) to study ‘how the presence of context would affect the identification of Mandarin tones and emotion tones’. The latter was referred to as the isolation condition. The authors did not offer any hypotheses or predictions for the effect of context based on current understanding of the effect of tonal coarticulation on tonal perception and that emotion expression can affect the prosody of the whole sentence.

It should also be pointed out that the isolation condition cannot be representative of how a lexical tone is produced in isolation, i.e. with no neighbouring sounds, or how an emption is expressed in a monosyllabic word with no neighbouring sounds. When a monosyllabic word is produced in isolation, it is produced as a one-word utterance, prosodically speaking. It has no tonal coarticulation and carries all the prosodic adjustments that need to be made to express a certain emotion.

Besides, in the ‘context’ condition, the target words were preceded by a Tone-1 word and followed by neutral tone-word in all sentences. Can the current results be generalised to other tonal contexts? For example, what happens when the target word is preceded by a Tone-2, Tone-3 or Tone-4 word?

g. Experiment 2, lines 609-615: The authors stated that ‘the order of the stimuli within each block was randomized for each participant’. How was this done exactly? Take the block of isolated syllables for tonal identification for example. What kind of principles were used in the randomization?

h. Experiment 2: I don’t understand why stimuli produced in the neutral emotion were not included in the task on perception of emotion. Excluding this option might potentially increase the chance of choosing the right emotion from 20% to 25%.

i. The discussion on the differences in the context and isolation conditions was somewhat confusing. The authors seemed to acknowledge that there were prosodic cues to emotion outside the target word in a sentence in some places (e.g. lines 829-830). But in the conclusion section, the authors attributed the better performance in the context condition to listeners’ knowledge of tonal co-articulation (lines 843-844). This is true for the perception of tones but not true for the perception of emotion.

2. Has the statistical analysis been performed appropriately and rigorously?

a. The data from Experiment 2 were categorical by nature: 4 categories for identification of lexical tones; 4 categories for identification of emotions. These data should thus be analysed using methods that can deal with categorical data. Repeated measures ANOVA, which was used in the paper, is not the appropriate analysis. The authors can consider using the mixed-model multinomial regression analysis in R.

b. The effect of emotion on the shape of the f0 contours of the lexical tones was discussed based on visual observations (lines 380-390). A more rigorous and more appropriate approach would be to use the Functional Data Analysis to mathematically quantify the shapes of each contour in terms of functional principle components and then use the fPCs as dependent variables.

Gubian, M., Torreira, F., & Boves, L. (2015). Using Functional Data Analysis for investigating multidimensional dynamic phonetic contrasts. Journal of Phonetics, 49(0), 16–40. https://doi.org/http://dx.doi.org/10.1016/j.wocn.2014.10.001

Chen, A., & Boves, L. (2018). What’s in a word: Sounding sarcastic in British English. Journal of the International Phonetic Association, 48(1). https://doi.org/10.1017/S0025100318000038

c. Crucial details are missing on the f0 contours presented in Figure 1: How were the f0 contours of the lexical tones in Figure 1 determined? Were the mean f0 contours of all speakers? If so, how were they extracted? Were they time-normalised?

d. Statistics on all the pairwise comparisons should be provided for each analysis.

3. Have the authors made all data underlying the findings in their manuscript fully available?

The authors stated that ‘All relevant data are within the manuscript and its Supporting Information files.’ I did not find out how to access Supporting Information in PLOS One. It seems that I could only download the manuscript. I ticked the ‘Yes’ box for this question but I’d like acknowledge that I do not have the information to judge on the availability of the data in Supporting Information that has not been presented in the manuscript.

4. Is the manuscript presented in an intelligible fashion and written in standard English?

I fear that the clarity of the manuscript has been affected by the presentation and exposition, even though there are very few language errors. The major issue I have with the writing is that the authors regularly made claims without substantiating them. A few examples:

a. The authors sketched four possible scenarios for how emotion may affect the production and perception of tones on pp. 4-5 (lines 70-86). These scenarios appear poorly informed given what has already been known.

b. Lines 98-99: ‘Despite the universal appearance of emotion tones, their acoustic correlates vary across studies.’ I do not understand claims such as this one that seem to contain contradictory information. In this particular case, the review of the literature in the preceding lines clearly pointed to cross-linguistic differences in how f0, duration, intensity are varied to express the same emotion.

c. Lines 167-169: These lines were supposed to summarise the main findings reviewed in the same paragraph. However, the preceding review did not talk about discrimination of consonants and recognition of spoken words. The findings reviewed were mostly concerned with recognizing word meaning.

d. Lines 175-184: These lines were supposed to give more details on Singh et al.’s work after the summarising sentence on what their work was about in lines 173-175. However, what was presented in lines 175-184 was hard to follow and did not really match the summarising sentence.

e. Lines 195-197 on the limitation of Liang and Chen’s study: “Since no acoustic analysis was reported, it is not clear to what extent these findings could be explained by the actual acoustic difference among the emotional tones.” I am not sure I can follow the logic here. Liang and Chen had the stimuli rated by a native speaker of Mandarin in terms of emotion before presenting them to their participants. The emotions were correctly recognised in all cases. This would allow the authors to conclude that emotions were accurately produced in the stimuli and differences found in the perception of tones in different emotions could be attributed to the differences in prosodic realisation of emotions.

f. Lines 231-232: “Although emotional tones appear to be perceived in a similar way irrespective of specific languages, it is not clear whether the use of lexical tones in tonal languages affects the perception of emotional tones.” Immediately following these lines, the authors reviewed Wang et al.’s study on exactly this topic. Their findings seem quite straightforward to me. What is still not clear and should therefore be investigated in the current study then?

g. Lines 271-273: I have the same concern as with lines 231-232.

h. Lines 256-259: “Evidence from L1 and L2 studies of segmental and tonal perception further suggests that tonal language experience affects how lexical tones and segmental structure are processed when they are produced with emotional tones.” Such studies were not reviewed in the preceding sections.

i. Lines 673-676: The description of the results (Tables 8, 9) was not accurate.

j. Lines 746-747: The description of the result was not accurate.

6. PLOS authors have the option to publish the peer review history of their article (what does this mean?). If published, this will include your full peer review and any attached files.

Reviewer #1: **Yes: **Liquan Liu

Reviewer #2: No

---

## [Author Response · Author response to Decision Letter 0]

28 Feb 2022

Editor comments:

Specifically, the reviewers acknowledge the merits of this study but see major problems with the integration with the current literature as well as with the justification of the design and the appropriateness of the analyses. Because PLOS ONE emphasises methodological soundness over novelty and impact I am minded to give you the opportunity to address the reviewers' comments very carefully. Reviewer 2 makes particularly pertinent and detailed comments that I encourage you to address. I particularly see the need to address the comments regarding ANOVA as an unsuitable statistical approach to these data. In addition, please make sure the data are being made available to the reviewers - you state that they are but the reviewers had trouble finding them. Finally, I would like to point out that request of revision does not constitute guarantee of acceptance of the revised version.

Response: We thank Dr. Kempe for the reminder regarding methodological soundness. We have followed the reviewers’ advice to revamp the conceptual foundation of the study. We have replaced the ANOVA with a mixed-model multinomial regression analysis for Experiment 2. We made sure the data is available to the reviewers.

Reviewer #1 comments:

The study is interesting and relevant. I enjoy reading it very much. The separation between acoustics and perception is wonderful. Please find the points of improvement/discussion below.

Response: We thank Dr. Liu for recognizing the merit of this study.

1. Is the manuscript technically sound, and do the data support the conclusions?

Narratives can appear more concisely throughout the manuscript.

Response: We have revised the manuscript to make it more concise.

If possible, a bit more discussion on the other way (lexical affecting emotional tone) would be appreciated.

Response: We have added more information about how lexical tones effect the acoustics and perception of emotions:

• In the section “How Emotions Affect Speech Acoustics”, we added two studies showing restricted F0 variations of emotions in other tonal languages (Chong, Kim, & Davis, 2015, for Cantonese; and Luksaneeyanawin, 1998, for Thai). We also added a Mandarin-Japanese study that showed the opposite pattern (Li, Jia, Fang, & Dang, 2013). 

• In the section “How Emotions Are Perceived in Speech”, we added discussion of a study on emotion recognition in Mandarin (Wang, Ding, & Gu, 2012). 

There is a potential design issue in the “context” condition, which can be resolved by careful discussion since the experiments have been done.

Response: We have clarified our use of the term “context” throughout the manuscript.

Based on the previous comment, please double-check the research questions. Make sure the questions asked are the variables being examined.

Response: We have revised the introduction to make sure the research questions are aligned with the variables being examined.

2. Has the statistical analysis been performed appropriately and rigorously?

The effect size is omitted in the report.

Response: We have included effect size as instructed.

3. Have the authors made all data underlying the findings in their manuscript fully available?

Maybe I missed this: it is unclear where (anonymous) data are stored.

Response: We apologize for missing the data in the original submission. As instructed, we have included the minimal data set.

4. Is the manuscript presented in an intelligible fashion and written in standard English?

The manuscript is understandable but is encouraged to be double-checked on its English writing style. There is space for further improvement.

Response: We have asked a native speaker of English to check our word choice, grammar, and writing style.

Detailed comments:

Abstract

1. Polish writing, including style, words and grammar

A. The starting sentence can be polished to enhance the significance of the study

Response: We have revised the starting sentence to enhance the significance of the study.

B. “the effect of emotional tones on Mandarin tone acoustics varied depending on specific Mandarin tones, specific emotional tones, and talker gender.” Please rephrase.

Response: We have rephrased this sentence to improve its clarity.

C. “select materials”: selected? (same in Ln283)

Response: We have changed the word here and elsewhere.

D. “Emotional tones also affected the four acoustic measures to different degrees.” Please integrate with the narratives prior.

Response: We have integrated this sentence into a previous sentence.

2. Reflecting on the narratives of Exp.2, clarify whether Exp.1 is presented in isolation or in context.

Response: We have clarified that acoustic analyses in Experiment 1 were conducted on target syllables extracted from a carrier phrase.

Introduction

Ln48: I understand that the authors adopted a four-basic emotion model. I wonder why the more commonly accepted six-basic emotion model, adding surprise and disgust in the study, is not considered.

Response: We thank Dr. Liu for pointing this out. The four basic emotion model was proposed by Ortony & Turner (1990) and the six basic emotion model was proposed by Ekman (1999). Both were cited extensively: Ortony & Turner (1990) was cited 3107 times and Ekman (1999) 4399 times. We decided to adopt the four emotion model because the four emotions are included in the six-emotion model, and most of the studies on Mandarin have used the four-emotion model (Yuan, Shen, & Chen, 2002; Zhang, Ching, & Kong, 2006; Lin & Fon, 2012; Wang & Qian, 2018; Chang, Young, Li, Chu, & Ho, 2018; Chang, Young, & Yuan, 2010; Wang, Lee, & Ma, 2016). Adopting the four-emotion model would make comparisons with previous studies more straightforward.

Ekman, P. Basic Emotions. In Dalgleish T, Power T, editors, The Handbook of Cognition and Emotion. Sussex: John Wiley & Sons, Ltd.; 1999. p. 45‐60

Ln51: Please double check "amplitude" or "intensity"

Response: We have replaced “intensity” with “amplitude” so that “amplitude” is used consistently throughout the manuscript.

Ln70: 1. I have the feeling that these scenarios are restricted to certain context/isolation, or when only key cues are involved. This is later discussed in Ln575. 2. I wonder if these are more like logical possibilities unless the authors argue that different scenarios may occur with different emotional-lexical tone combos.

Response: We agree that the scenarios are more like logical possibilities, which did not contribute much to motivating the research questions. This point is also made by Reviewer 2. We have removed this paragraph and subsequent discussions that were based on this taxonomy.

Ln85: I wonder whether the other way around is worth highlighting, namely, how lexical tone affects emotion (e.g., emotional tone) perception? This is analysed and discussed after all (e.g., Ln805, later asymmetry)

Response: As noted in our earlier response, we have added more information regarding the effect of lexical tone on emotion perception to the section “how emotions are perceived in speech”.

Ln192: Does falling tone affect anger perception?

Response: Liang and Chen (2019) examined Mandarin tone perception but not emotional perception. Results from our Experiment 2 showed that anger perception in isolated syllables was more accurate in the falling tone (T4: 93%) compared to other tones (T1: 77%; T2: 76%; T3: 53%). Anger perception in syllables embedded in the carrier phrase was comparable among the four tones (T1: 97%; T2: 97%; T3: 96%, T4: 99%).

Ln198-212: Some information here seems to have been mentioned earlier.

Response: This paragraph was intended to offer a summary of the studies reviewed in this section. In light of the reviewer’s comment, we have deleted this summary.

Ln232, 234: Wang et al. - repetitive presentation

Response: We have rewritten this section to remove the repetition.

Ln261: It is unclear why lexical tones lacking F0 variation would be predicted to make the emotional tone less perceptible in the first place. Can it be the other way around? That is, since lexical tone has no variation, would variations of emotion prevail?

Response: We thank Dr. Liu for this insight. This statement was intended to be a summary of Wang et al. (2015, 2018) reviewed earlier. We have rewritten this section to include both the authors’ hypothesis and the reviewer’s interpretation.

Experiment/Analyses

Ln310: Some tables are a little bit difficult to process although the meaning is clear. I do not have suggestions here because the authors have already given careful thoughts on the presentation of these tables.

Response: We thank Dr. Liu for understanding our effort to summarize the wide range of findings in the literature. We have added more information to the caption of this table and others to clarify the meaning of the symbols and notations.

Ln318: Why is 4 a good number of participants as the baseline for the acoustic characteristics of emotional tone?

Response: Ideally, the more participants, the better. However, our funding only allowed us to recruit a maximum of eight (4 female and 4 male) professional actors. To our knowledge, this number is comparable to those used in similar studies.

Ln537-540: Please rephrase to reduce repetitiveness.

Response: We have rewritten this section to remove the repetition.

Ln566: Please rephrase "In addition, we also"

Response: We removed “also” from this sentence as instructed.

Discussion

Regarding "Context": There can be several possibilities to understand this word in the “context” of the current paper. It may be worthwhile specifying which context the authors are referring to (carrier sentences).

Also, what the context adds to/change the picture needs to be spelt out more overtly (e.g., ease of semantic parsing and/or tonal coarticulation? If one, why not the other?)

One methodological issue is to what extent the discrepancy between context emotion (no emotion if I understand correctly) and target word emotion would affect the current findings in the context condition. This needs to be discussed.

Response: We thank Dr. Liu for this important observation. Reviewer 2 also made a similar comment. We have clarified what we mean by “context” throughout the manuscript.

Reviewer #2 comments: 

1. Is the manuscript technically sound, and do the data support the conclusions?

a. The current study vs. directly relevant past studies: This study is a potentially very valuable addition to the line of research on the influence of emotion on the production and perception of lexical tones in Mandarin. Different from past studies, the present study looks at the influence of emotion on lexical tones in both production and perception. However, this does not mean that past work with a focus on either the production side of the coin or the perception should be disconnected from the current study. In other words, the authors did not seem to make a sufficient use of existent findings to motivate their study and develop the hypotheses and predictions. Specially, the authors focused on how the realisation of each of the five basic emotions impacts the f0 mean, f0 range, intensity and duration of the target syllable. The literature review 'how emotional tones affect speech acoustics' only discussed the differences in the degree to which an acoustic parameter is exploited to express emotions between tone and non-tone languages. The authors referred to 'the literature reviewed' when presenting their predictions for changes in f0, duration and intensity (lines 300-302). But past studies on how emotion expression influences the f0 and duration of each lexical tone in Mandarin was not reviewed. Furthermore, although research on the effect of emotion expression on the perception of Mandarin tones by native speakers of Mandarin and second language learners of Mandarin is small in numbers, there has been some published work on this topic, e.g. Liang & Chen (2019), which was published in the ICPhS 2019 proceedings. This work was mistakenly cited as Liang and Chen's MA thesis in the references section, summarised in the section 'How emotional tones affect speech perception' and subsequently dismissed. This is a pity because the authors could have built their work on the findings from Liang and Chen, especially considering the fact that they had the same research question on the production side as Liang and Chen, and used a similar method in both Experiment 1 and Experiment 2 to Liang and Chen’s methodology. The main differences concerning production elicitation seems to be that Liang and Chen used phototactically legal non-words as target words and systematically varied the tonal contexts preceding the monosyllabic target words whereas the current study used real words and had only one tonal context, i.e. the monosyllabic target words were preceded by tone 1 in the carried sentence. Liang and Chen's findings can thus provide a very useful starting point for the current study in terms of both hypothesis formulation and methodology.

Response: We thank the reviewer for this important advice on grounding the current study in the literature. We agree that the study by Liang and Chen (2019) offers an excellent foundation for this study. We have revised the introduction, results, and discussion to highlight how Liang and Chen’s (2019) study has been used to develop our research questions and hypotheses, and how our results compared to Liang and Chen’s (2019) findings.

We also agree that a more meaningful connection should be made with previous studies focusing on either production or perception. We have rewritten the introduction and discussion to enhance the connection.

Re: “But past studies on how emotion expression influences the f0 and duration of each lexical tone in Mandarin was not reviewed.” Thank you for the comment. We have added a summary of Chao’s (1933) description of how emotions are implemented in Mandarin tones. We have also included a discussion of Li, Fang, and Dang’s (2011) acoustic study. 

Finally, we apologize for the confusion between Liang and Chen’s ICPhS proceedings and Liang’s MA thesis. We had referenced the thesis in the original version because the thesis seemed to provide more details of the study. In light of the reviewer’s feedback, we have used the ICPhS citation throughout the manuscript.

b. Taiwan Mandarin is prosodically not the same as Beijing Mandarin (e.g. in prosodic realisation of focus), even though they both have the same four lexical tones. This difference is not sufficiently acknowledged throughout the study. Notably, the similarities in findings between this study and those from Liang and Chen (2019) can have interesting implications for the prosodic similarities and differences regarding the interaction between emotion and lexical tones between Taiwan Mandarin and Beijing Mandarin.

Response: We thank the reviewer for bringing to our attention this comparison. We have acknowledged this difference as a limitation of the current study.

c. Experiment 1: The participants were specifically instructed not to place 'excessive stress on the target syllables'. This seems to suggest that the participants have no uniformed way to determine the information structure of each target sentence. They could have treated each target sentence as a response to a 'what-happens'-like question, rendering the whole sentence focal (broad focus) or treated the target word as the focus (narrow focus) in spite of the instructions. What the participants did regarding the information structure of the target sentences may influence how they used prosody to realise emotion and tones. For example, they might have distributed the prosodic changes over the whole sentence if they had a broad-focus analysis, or the prosodic changes might have concentrated in the target syllables if they had a narrow-focus analysis.

Response: We thank the reviewer for this insight. Our intention was to elicit a broad focus, i.e., distributing the prosodic changes over the whole sentence, instead of narrowly focusing on the target word. We instructed the participants to avoid placing excessive stress on the target word because the carrier phrase was the same for different target words, and we worried there would be a tendency for the speakers to emphasize the target word. We have revised the description to clarify that the instruction was intended to elicit a broad-focus analysis.

d. Experiment 1: The recordings from the participants were rated by 30 native speakers of Taiwan Mandarin. No information was given on who these participants were and how they were recruited. Were they the same participants who did Experiment 2?

Response: Yes, the same participants also participated in Experiment 2. We apologize for not specifying this information in the original manuscript. We have added this information in the revision. Also, there were actually 36 participants for the rating task. We apologize for the error. We have made corrections throughout the manuscript.

e. Experiment 2: Experiment 2 hinged on two opposing assumptions. Neither is well motivated. One of the assumptions was that ‘emotional tones alter the canonical acoustic characteristics of Mandarin tones, making it difficult for listeners to retrieve the Mandarin tones’ (Lines 554-556). The authors did not specify anywhere in the manuscript what the ‘canonical acoustic characteristics of Mandarin tones’ should be. I do not think that past studies have provided evidence for such a claim either. Instead, it has been shown that the changes in the acoustic realisation of lexical tones related to the expression of emotions are restricted to the gradient changes in terms of higher of lower mean pitch, smaller or wider pitch range, or longer or shorter duration. Such changes are phonetic by nature because lexical tones are not defined by absolute pitch, duration and intensity. The other assumption is that ‘listeners are able to disaggregate the acoustic changes into two sources – Mandarin tones and emotional tones’. This assumption is itself based on the assumptions that lexical tones have absolute f0, duration and intensity values, which is not the case, and the perception of emotion prosody is categorical, for which evidence has emerged (e.g. Laukka 2005).

Laukka, P. (2005). Categorical perception of vocal emotion expressions. Emotion, 5(3), 277–295.

Response: We thank the reviewer for these observations. We certainly agree that lexical tones are not defined by absolute pitch, intensity, or duration. The acoustic characteristics of tones vary across talkers, phonetic contexts, and emotions. Our use of the term “canonical” was meant to highlight that emotions, like talkers (e.g., Moore and Jongman, 1997), tonal contexts (e.g., Xu, 1997), and other sources of acoustic variability, may change the citation form of lexical tones. The gradient acoustic changes noted by the reviewer illustrate exactly the idea. It was not our intention to claim that a lexical tone is associated with an ideal set of acoustic characteristics. In light of the reviewer’s comment, we have removed the term “canonical” and rephrased this statement to avoid that impression.

Our use of the term “disaggregate” was intended to highlight that the listener’s task is to interpret the acoustic signal by taking into consideration possible sources of acoustic variability. That is, listeners need to analyze the acoustic signal in terms of lexical tones, emotional tones, and other sources that contribute to the acoustic variability. This is analogous to interpreting lexical tones by considering talker F0 range (e.g., Moore and Jongman, 1997). It was not our intention to assume a lexical tone is associated with an ideal set of acoustic characteristics. We have rephrased this statement to avoid that impression.

We thank the reviewer for bringing Laukka (2005) to our attention. Since our study was not designed to test whether emotions are perceived categorically or not, we did not include the study in the references. 

f. The factors ‘context’ vs. ‘isolation’ in Experiment 2: The target words were presented to the participants in both their original carrier sentence and out of their original carrier sentence (e.g. cut out of the carrier sentences) to study ‘how the presence of context would affect the identification of Mandarin tones and emotion tones’. The latter was referred to as the isolation condition. The authors did not offer any hypotheses or predictions for the effect of context based on current understanding of the effect of tonal coarticulation on tonal perception and that emotion expression can affect the prosody of the whole sentence.

It should also be pointed out that the isolation condition cannot be representative of how a lexical tone is produced in isolation, i.e. with no neighbouring sounds, or how an emption is expressed in a monosyllabic word with no neighbouring sounds. When a monosyllabic word is produced in isolation, it is produced as a one-word utterance, prosodically speaking. It has no tonal coarticulation and carries all the prosodic adjustments that need to be made to express a certain emotion.

Besides, in the ‘context’ condition, the target words were preceded by a Tone-1 word and followed by neutral tone-word in all sentences. Can the current results be generalised to other tonal contexts? For example, what happens when the target word is preceded by a Tone-2, Tone-3 or Tone-4 word?

Response: We thank the reviewer for this important observation. Reviewer 1 also made a similar comment. We apologize for not specifying the hypotheses regarding context in the original version. We have elaborated on our characterization of context throughout the manuscript. We have also added specific hypotheses to the section “the present study”.

We agree that the “isolation” condition does not represent how a lexical tone is produced in isolation. As the reviewer correctly pointed out, the “isolated” syllables were not produced in isolation; rather, they were excised from a sentence with the carrier phrase. We have revised the manuscript to clarify this and the nature of the isolation vs. context comparison.

We agree with the reviewer’s observation about generalization. Since the carrier phrase was the same for all target syllables, our results cannot generalize to other tonal contexts. On the other hand, Liang and Chen (2019) systematically manipulated the preceding tone. We have revised the manuscript to acknowledge this limitation.

g. Experiment 2, lines 609-615: The authors stated that ‘the order of the stimuli within each block was randomized for each participant’. How was this done exactly? Take the block of isolated syllables for tonal identification for example. What kind of principles were used in the randomization?

Response: We used the Random Number Generator in LabVIEW (National Instruments) for randomization. Each of the 120 syllables was assigned a number ranging from 1 to 120. The Random Number Generator was then used to create a unique presentation order for each participant. We have added this description to specify the randomization procedure.

h. Experiment 2: I don’t understand why stimuli produced in the neutral emotion were not included in the task on perception of emotion. Excluding this option might potentially increase the chance of choosing the right emotion from 20% to 25%.

Response: Thank you for this observation. We had intended to have 4 response options for both Mandarin tone identification (4 tones) and emotion recognition (4 emotions), but we agree we should have included the neutral emotion in the emotion recognition task. We have acknowledged this limitation in the manuscript. 

i. The discussion on the differences in the context and isolation conditions was somewhat confusing. The authors seemed to acknowledge that there were prosodic cues to emotion outside the target word in a sentence in some places (e.g. lines 829-830). But in the conclusion section, the authors attributed the better performance in the context condition to listeners’ knowledge of tonal co-articulation (lines 843-844). This is true for the perception of tones but not true for the perception of emotion.

Response: We thank the reviewer for this observation. As acknowledged earlier, context was indeed not explained sufficiently in the original version. We have revised the manuscript to clarify the interpretation of context. 

2. Has the statistical analysis been performed appropriately and rigorously?

a. The data from Experiment 2 were categorical by nature: 4 categories for identification of lexical tones; 4 categories for identification of emotions. These data should thus be analysed using methods that can deal with categorical data. Repeated measures ANOVA, which was used in the paper, is not the appropriate analysis. The authors can consider using the mixed-model multinomial regression analysis in R.

Response: We thank the reviewer for pointing this out. As instructed, we have redone the analysis by using a mixed-model multinomial regression analysis in R. We have rewritten the results and discussion based on the updated analysis.

b. The effect of emotion on the shape of the f0 contours of the lexical tones was discussed based on visual observations (lines 380-390). A more rigorous and more appropriate approach would be to use the Functional Data Analysis to mathematically quantify the shapes of each contour in terms of functional principle components and then use the fPCs as dependent variables.

Gubian, M., Torreira, F., & Boves, L. (2015). Using Functional Data Analysis for investigating multidimensional dynamic phonetic contrasts. Journal of Phonetics, 49(0), 16–40. https://doi.org/http://dx.doi.org/10.1016/j.wocn.2014.10.001

Chen, A., & Boves, L. (2018). What’s in a word: Sounding sarcastic in British English. Journal of the International Phonetic Association, 48(1). https://doi.org/10.1017/S0025100318000038

Response: We thank the reviewer for pointing this out and offering the references. We agree that the Functional Data Analysis would be the most rigorous approach for a quantitative analysis of dynamic F0 contours. However, the focus of our acoustic analysis was the four measures (mean F0, F0 range, mean amplitude, and duration), not the F0 contours. Our purpose of including the figure and qualitative description was to show that the four tones were produced as intended. We certainly agree that analyzing the dynamic F0 contours would be quite informative. However, we feel it is best to leave the analysis to a separate study because including such an analysis would make our paper even longer than it already is. 

Finally, as the reviewer pointed out earlier, the acoustic realization of lexical tones in the expression of emotions is typically restricted to gradient changes in terms of mean pitch, pitch range, or duration. We agree with the reviewer that these acoustic measures would offer more insights for the purpose of this study. We have revised this section to acknowledge that our observations are qualitative in nature. We have also included the reference offered by the reviewer.

c. Crucial details are missing on the f0 contours presented in Figure 1: How were the f0 contours of the lexical tones in Figure 1 determined? Were the mean f0 contours of all speakers? If so, how were they extracted? Were they time-normalised?

Response: We apologize for missing these details. We have added a description of how the F0 contours were determined. Each F0 contour was indeed generated by averaging over all speakers of the same sex. The contours were also time-normalized as the reviewer pointed out. 

d. Statistics on all the pairwise comparisons should be provided for each analysis.

Response: We have made sure that all pairwise comparisons are provided where applicable.

3. Have the authors made all data underlying the findings in their manuscript fully available?

The authors stated that ‘All relevant data are within the manuscript and its Supporting Information files.’ I did not find out how to access Supporting Information in PLOS One. It seems that I could only download the manuscript. I ticked the ‘Yes’ box for this question but I’d like acknowledge that I do not have the information to judge on the availability of the data in Supporting Information that has not been presented in the manuscript.

Response: We apologize for missing the data in the original submission. We have supplied a minimal data set as instructed.

4. Is the manuscript presented in an intelligible fashion and written in standard English?

I fear that the clarity of the manuscript has been affected by the presentation and exposition, even though there are very few language errors. The major issue I have with the writing is that the authors regularly made claims without substantiating them. A few examples:

a. The authors sketched four possible scenarios for how emotion may affect the production and perception of tones on pp. 4-5 (lines 70-86). These scenarios appear poorly informed given what has already been known.

Response: We agree that the four scenarios were not grounded in the literature and did not contribute to motivating the research questions. Reviewer 1 also made a similar comment. We have removed this paragraph and subsequent discussions that were based on this taxonomy. Instead, we have followed the reviewer’s suggestion to use Liang and Chen (2019) as the foundation of this study.

b. Lines 98-99: ‘Despite the universal appearance of emotion tones, their acoustic correlates vary across studies.’ I do not understand claims such as this one that seem to contain contradictory information. In this particular case, the review of the literature in the preceding lines clearly pointed to cross-linguistic differences in how f0, duration, intensity are varied to express the same emotion.

Response: We apologize for the confusion. We have rephrased this statement to resolve the contradiction.

c. Lines 167-169: These lines were supposed to summarise the main findings reviewed in the same paragraph. However, the preceding review did not talk about discrimination of consonants and recognition of spoken words. The findings reviewed were mostly concerned with recognizing word meaning.

Response: By “discrimination of consonants” we were referring to Mullenix et al. (2002), who manipulated the final consonant of the names (e.g., Todd-Tom). By “recognition of spoken words”, we were referring to Mullenix et al. (2002), Kitayama and Ishii (2002), Ishii et al. (2003), Nygaard and Lunders (2002), Nygaard and Queen (2008): they all used some form of a word recognition task. This summary is no longer in the revised manuscript due to extensive rewriting.

d. Lines 175-184: These lines were supposed to give more details on Singh et al.’s work after the summarising sentence on what their work was about in lines 173-175. However, what was presented in lines 175-184 was hard to follow and did not really match the summarising sentence.

Response: We apologize for the lack of clarity. We have rewritten the description to clarify what was done in this study.

e. Lines 195-197 on the limitation of Liang and Chen’s study: “Since no acoustic analysis was reported, it is not clear to what extent these findings could be explained by the actual acoustic difference among the emotional tones.” I am not sure I can follow the logic here. Liang and Chen had the stimuli rated by a native speaker of Mandarin in terms of emotion before presenting them to their participants. The emotions were correctly recognised in all cases. This would allow the authors to conclude that emotions were accurately produced in the stimuli and differences found in the perception of tones in different emotions could be attributed to the differences in prosodic realisation of emotions.

Response: We thank the reviewer for this observation. We agree that ratings by native speakers constitute valid evidence that emotions are produced as intended. We were simply pointing out that no acoustic analysis was reported in the study. We have removed this statement to avoid the impression that the stimuli were not valid.

f. Lines 231-232: “Although emotional tones appear to be perceived in a similar way irrespective of specific languages, it is not clear whether the use of lexical tones in tonal languages affects the perception of emotional tones.” Immediately following these lines, the authors reviewed Wang et al.’s study on exactly this topic. Their findings seem quite straightforward to me. What is still not clear and should therefore be investigated in the current study then?

Response: We apologize for the confusion. We have revised this statement to resolve the contradiction.

g. Lines 271-273: I have the same concern as with lines 231-232.

Response: We apologize for the confusion. We have removed this sentence from the revision.

h. Lines 256-259: “Evidence from L1 and L2 studies of segmental and tonal perception further suggests that tonal language experience affects how lexical tones and segmental structure are processed when they are produced with emotional tones.” Such studies were not reviewed in the preceding sections.

Response: This statement was a summary of Singh, Lee, and Goh (2016), and Liang and Chen (2019). Both studies showed that tonal language experience affected perception of speech produced with emotions. This summary is no longer in the revised manuscript because of extensive rewriting.

i. Lines 673-676: The description of the results (Tables 8, 9) was not accurate.

Response: We have revised these statements to make sure the description is accurate.

j. Lines 746-747: The description of the result was not accurate.

Response: We have revised these statements to make sure the description is accurate.

---

## [Decision Letter · Decision Letter 1]

6 Jun 2022

PONE-D-21-23777R1Emotional tones of voice affect the acoustics and perception of Mandarin tonesPLOS ONE

Dear Dr. Chu,

Thank you again for submitting your revised manuscript to PLOS ONE. Let me start by reiterating my apologies for the delay which were due to some difficulties with recruiting reviewers for the revised version.  As you will see, both reviewers acknowledge the improvements that you have carried out in response to the first round of reviews and have taken the previous reviews into account in appraising your current revision. It is also clear that there is considerable overlap in the reviewers' remarks, especially pertaining to better justification of the four-emotion-model and potential repercussions of this choice and to improvements in the statistical analyses. In addition, the reviewers have made a number of further insightful suggestions that I urge you to consider. Even though one reviewer categorises the required work as 'Minor Revision' I feel that there are still some more substantial improvements required, which is why I decided to return it as a 'Major Revision'. I hope that the reviewers' comments will aid you in a subsequent revision of this submission.

We look forward to receiving your revised manuscript.

Kind regards,

Vera Kempe

Academic Editor

PLOS ONE

Reviewers' comments:

Reviewer's Responses to Questions

**Comments to the Author**

1. If the authors have adequately addressed your comments raised in a previous round of review and you feel that this manuscript is now acceptable for publication, you may indicate that here to bypass the “Comments to the Author” section, enter your conflict of interest statement in the “Confidential to Editor” section, and submit your "Accept" recommendation.

Reviewer #3: (No Response)

Reviewer #4: (No Response)

2. Is the manuscript technically sound, and do the data support the conclusions?

Reviewer #3: Partly

Reviewer #4: Partly

3. Has the statistical analysis been performed appropriately and rigorously? 

Reviewer #3: No

Reviewer #4: No

4. Have the authors made all data underlying the findings in their manuscript fully available?

Reviewer #3: No

Reviewer #4: Yes

5. Is the manuscript presented in an intelligible fashion and written in standard English?

Reviewer #3: Yes

Reviewer #4: Yes

6. Review Comments to the Author

Reviewer #3: In line with the previous two reviewer’s general comments, I agree that the current work is relevant and can be a valuable addition to our current knowledge base.

While the authors have either acknowledged as limitations or addressed the majority of the points raised, I feel that there are still a number of issues that have not been satisfactorily resolved.

In the course of this review, I have noted a number of additional issues beyond what was raised in the previous review. Consequently, I felt that the current manuscript still lacks a strong rationale and justification for some of the decisions made regarding the study design and this has implications on the interpretability of the results.

1. Is the manuscript technically sound, and do the data support the conclusions?

A) Reviewers 1 and 2 both made similar remarks about the disconnect between the broader motivations of the study under review and how it is positioned within the context of similar research. Reviewer 2 recommended that the authors use the work of Liang and Chen (2019) as an anchor point for the current manuscript.

The authors have done well to incorporate the reviewers comments and the Liang and Chen (2019) study is given prominence in the introduction.

However, given the importance of the study as the cornerstone of the current work, this manuscript will benefit from a revision to provide greater clarity and details of the Liang and Chen study. In its current form, I feel that a reader will need to first review Liang and Chen’s thesis in order to follow this manuscript.

B) Reviewer 1 asked why the authors adopted a four basic emotion model (angry, happy, fear and sad) instead of a six emotion model (which on top of the four aforementioned emotion types, also include surprise and disgust).

The authors replied that the decision was made on the basis that most studies on Mandarin have adopted the four emotion model, hence using the four emotion model would make comparisons with previous studies more straightforward.

I am not sure if I quite follow the logic of this reasoning. As the four emotion model is a subset of the six emotion model, straightforward comparisons can still be made if the current study adopted the six emotion model.

Moreover, using the six emotion model has a number of other benefits. One, it would allow comparisons against a much larger literature base (studies using either four or six emotion models) and potentially against the wider literature of other tonal and non-tonal languages.

The second benefit pertains to Reviewer 2’s comment about the inclusion of Neutral as a possible response option for Experiment 2 to reduce chance level accuracy (from 25% to 20%). A six emotion model would achieve a similar effect by having two additional emotion types as response options.

A third advantage, which I feel is more critical, relates to the aim of study. Fundamental Frequency (F0) is considered to be one of the most salient carriers of emotion information as it is argued that F0 may be an acoustic correlate of arousal (Scherer, 2003). For example, emotions with high arousal such as anger may be expressed with a higher mean F0 and conversely, emotions with low arousal such as sadness may be expressed with a lower mean F0. In this regard, of the four emotions examined in this study, only one (Sad) is considered to be of low arousal and the study may therefore benefit from the inclusion of a larger range of emotion types of varying arousal levels to allow a more nuanced study of the interaction between emotions and lexical tone production.

C) Reviewer 2 raised a question regarding the randomisation procedure that was undertaken within each experimental block in Experiment 2. Similarly, I have a concern regarding randomisation but of the experimental blocks. From my reading, I understand that the order of block presentation was fixed for all participants such that they always viewed stimuli in the isolation condition before the context condition.

I am curious about the authors reasoning for this. I feel that this is may be a weakness as confounds such as training or exposure may have an unwanted impact on the observed results (e.g., participants have familiarised themselves with the stimuli by the time they get to the context condition, hence resulting in higher identification accuracy). This is given greater scrutiny especially since the authors are making claims based on the comparison of the isolation to the carrier condition.

D) Reviewer 2 asked why Neutral stimuli were not included in the emotion perception task as the exclusion of this option had raised chance level accuracy rates from 20% to 25%.

The authors responded on page 33, third sentence from the top of the page (pardon the clunky referencing but there doesn’t seem to be line numbering on the revised version), that ‘Neutral was regrettably not included’. I think it would be more meaningful to justify why Neutral was not included, rather than an admission that it was not there.

Nevertheless, in my opinion, the exclusion of Neutral has the benefit of making this a forced choice task that prevented the potential abuse of Neutral as a bin for stimuli that participants were uncertain about.

However, my concern is about why Neutral as a response option was excluded when Neutral stimuli were presented to the participants in the emotion identification task. From my reading, it appears that Neutral stimuli were presented in every experimental block (each experimental block consists of 120 stimuli, 3 syllables, 4 tones, 5 emotions, and 2 repetitions – bottom of pg. 31). Does this also mean that the participants would be giving an incorrect response to the Neutral stimuli by default simply because the Neutral option was not available?

E) Both reviewers had raised issues regarding the unclear rationale behind the isolation vs. context comparisons, and the lack of clarity in the discussion of the findings.

Unfortunately, despite the revision, I am unsure if I follow the logic and significance of the comparison between the isolation and context conditions. There were no predictions or references given that would contradict the rather intuitive hypothesis and critical discussions regarding these findings still seems a little thin.

This section could potentially benefit from another rewrite to clearly state the significance of the isolation vs. context comparison, the underlying or competing mechanisms at work, and how this comparison adds to our current knowledge base. In particular, I find the second paragraph of page 29 hard to follow.

F) Another point that is unclear to me is the motivation behind the authors’ aim to examine sex differences. From my reading, there is only one justification given, which is to extend Liang and Chen’s (2019) study where all stimuli were produced by a single female speaker (this information was not given in the current manuscript).

Despite the focus on sex differences, it is unclear whether the analyses here were meant to be exploratory or confirmatory. The manuscript does not provide adequate references or predictions on precisely how sex may have an impact on lexical and emotion tone production and why it is important that we study this.

I note that this issue was not raised by the previous reviewers. My reason for bringing this up is that, the manuscript will be easier to read if clearer objectives and deeper insights into the authors’ thought process is given. Moreover, this has implications regarding the appropriateness of the statistical tests undertaken by the authors.

From my understanding, at the broadest level, this study aims to address how emotion (not emotion and sex) affects the production and perception of Mandarin lexical tones. While it is interesting to tease apart and comment on sex differences, I feel that the logical next step, which is missing, is how these effects may be controlled for in order to draw inferences regarding the generalisability of the findings and its robustness against sex and other potential confounds arising from idiosyncratic individual differences.

In this regard, I highly recommend the use of mixed effects models where speaker, sex and perhaps even syllable type may be entered as random intercepts and slopes. This is aligned with Reviewer 2’s suggestion that mixed effects models be applied on data from Experiment 2. I am of the opinion that this should also be applied to Experiment 1 and with more rigour than what is currently done (more on this in the next section).

2. Has the statistical analysis been performed appropriately and rigorously?

A. Reviewer 2 recommended the use of Functional Data Analysis for the analysis of F0 contours.

The authors agreed that the use of Functional Data Analysis was appropriate but unwarranted as the current analysis of contours was intended to be a manipulation check of sorts to show that the four tones were produced as intended.

I agree that the use of Functional Data Analysis may not be necessary if the analysis was intended to be a simple visual sanity check. However, in its current form, explorations of the contours appear to be beyond the level of a simple sanity check as contrasts between sex and emotions were made. It is therefore inaccurate for the authors to claim that the ‘qualitative observations were meant to corroborate that the tones were ‘produced as intended’ (second last sentence of first paragraph on page 19) as it is not clear what the ‘intended’ shape of the contour is for the different sexes and emotion types.

My recommendation is for the authors to acknowledge that this is an exploratory examination to visualise and better understand the stimuli produced in the current study, rather than framing this as a comparison against a ‘standard’.

B. Reviewer 2 suggest that that mixed-model multinomial regression analysis in R may be a more appropriate analysis than the repeated measures ANOVA performed by the authors.

The authors have now applied mixed model logistic regression in their analysis of the data from Experiment 2.

More critically, I am concerned about how the variables were entered into the model in the revised manuscript. I am not sure if I understand why participants were entered as a random effect instead of variables such as sex, talker or syllable. It also unclear why the model was needlessly overcomplicated by the inclusion of the interaction effect between emotion and context when this is not the aim of the study (and also subsequently ignored by the authors – pg. 39). It would also be instructive for the authors to cite the R package that was used.

The significant interactions effects were further illustrated by the authors through the use of confusion matrices. While the findings presented are sound, I feel that an edit would provide better clarity. For example, the authors wrote ‘The most common error for Mandarin tones produced with ANGRY was Tone 4’. My suggestions would be to rephrase this to something along the lines of ‘There appears to be a response bias where Mandarin lexical tones, Tones 1 and 3 in particular, were most commonly misidentified as Tone 4 when presented in an Angry tone of voice’.

C. I would also like the authors to review how the confidence interval error bars of Figures 3 and 6 were generated. Some of the CI bars are in the negative range, which although can happen through calculation but not in actual measure (e.g., F0 range). This is likely indicative of a rather unusual or problematic distribution in the data which calls into question whether sufficient data preparation and cleaning (outliers?) has been conducted prior to analysis.

D. When conducting regression analysis, it is good practice to include within the body of the results section the statistics of the analysis such as the Beta estimates, standard errors, and Z-ratio. These are currently only available as downloadable supplementary information.

3. Have the authors made all data underlying the findings in their manuscript fully available?

A. Following the provided link, I am able to view only a simplified output of the mixed effects model. No other data was found.

4. Is the manuscript presented in intelligible fashion and written in standard English?

A. In general, both reviewers commented to the effect that edits were required to improve the quality of the manuscript. Reviewer 2 further noted that the authors frequently made unsubstantiated claims.

The authors have made extensive rewrites with assistance from a native speaker of English who provided feedback on word choice, grammar and writing style.

It is clear that the quality of the writing has improved in the revised manuscript on account of the substantial effort put in by the authors’ and their openness to suggestions. However, it can definitely benefit from another round of polishing and editing as there are areas within the manuscript (e.g., the section on confusion matrices) that although is grammatically sound, lacks clarity.

Reviewer #4: Review of "Emotional tones of voice affect the acoustics and perception of Mandarin tones"

Martijn Goudbeek, Tilburg University

This manuscript addresses and interesting question, namely how linguist and paralinguistic cues mutually influence each other in communication. Specifically, it does this by investigating how the expression of emotion in Mandarin affects the prouction and perception of tones 1 to 4 and, conversely, how the production of tones 1 to 4 affects the production and perception of emotion. A tonal language such as Mandarin provides an excellent testbed for the linguistics/paralinguistics interface, because emotional expression often modulates pitch, and pitch is one of the defining features (if not the defining feature) of Mandarin tones.

Since I did not review the orignal manuscript, I tried to pay close attention to the original comments and the replies from the authors. However, even though I did take the original reviews as a basis, there are some new remarks and recommendations in my review.

Theoretical points

------------------

One of the points of discussion in the original reviews was how to defend the use of four emotion categories. The authors have explained that these four are the four basic emotions in Ortony and Turners 1990 paper. This is somewhat unfortunate, since this paper is a highly cited /critique/ of the whole idea of basic emotion theory (a critique deemed relevant enough for Paul Ekman to directly engage the paper in his 1992 paper). That said, and although I would appreciate a better motivation for the -small number of- emotions included in the corpus, I do not particularly mind that there are only four. The more fundamental problem is that the low number of emotions (and the ones chosen) are never discussed in terms of the consequences for recognition in the vocal domain. For example, the fact that there is only one positive emotion (happiness) and only one low aroused emotion (sadness) strongly influences the decision problem that participants face. Likewise, it also strongly influences the results of the analysis, since (for example) the acoustic profile of sadness is so much different from the others in intensity, that significant results are bound to emerge. Contrast this with a situation where other low aroused emotions such as disgust or tenderness are in the dataset and you get very different results for, say, F0 or intensity. For a balanced and transparent discussion of the meaning of the results, factors like these needs to be taken into account, both in introducing the study and in the discussion section.

Along these lines I think the exclusive focus on basic emotions is needlessly limiting. Especially considering the fact that results and conclusions are often cast in terms of positive versus negative emotions, I think the dimensional approach to emotions (e.g., Russel, 1980, Russell and Feldman Barrett, 1999 and further) should be given proper attention (for exmaple, when describing the emotoins in the study, but also when interpreting the results, which is, as mentioned often already done in terms of valence -which is notably not a property of basic emotion theory which considers all emotions orthogonal categories). A reference to Laukka 2005 (in the rebuttal letter) is somewhat misleading, since although it provides evidence for categorical (which is not the samex`) perception of emotions, a paper from the same author in the same year using the same dataset uses the dimensional approach (Laukka, Juslin, & Bresin, 2005).

Methodolical / Statistical points

In addition to these theoretical considerations, there are some methodological issues that need to be addressed.

For the analysis of variance in experiment 1, the (statistical) design is somewhat unclear. The design is introduced with "For each of the four acoustic measures, a three-way mixed-design analysis of variance (ANOVA) was conducted [...] with emotion [] and Mandarin tones [...] as within-subject factors and talker sex [...] as a between-subject factor." What is not clear is why tone and emotion are within factors, but why other simulus characteristics that were deemed relevant in the construction of the corpus (syllable, repetition) were not used as within factors. Statistically, this variance is unexplained variance, but in a more sophiticated analysis, these could be random effects over which the study could generalize (see Judd, Westfall, and Kenny, 2012, but also Winter and Grice, 2021). In any case, the precise number of items in the analysis should be clarified, if only because otherwise the degrees of freedom in the analysis become difficult to interpret.

As a follow up to the same analysis (the mixed ANOVA of experiment 1), the authors use LSD as a post hoc test. This test does not correct in any way for multiple comparisons, thus increasing the possibility of a type I error. While there might be arguments to not correcting every multiple comparison with a bonferroni test, the choice to not correct at al (certainly with such a large number of comparisons. This is particularly relevant in light of Figure 1, where almost all CI's overlap substantially (indicating the lack of a significant difference) which is in stark contrast to the findings of the post hoc analysis.

For the second analysis, multinomial mixed effects models are indeed the correct way to analyze this kind of data. However, mixed models also enable the inclusion of items (in addition to participants) as a random effect (again, see Judd, Westfall, and Kenny, 2012, but many others). This appears to not have been done, severely limiting the generalizability of the findings. For the revision, items should be entered as a random effect in the analysis.

It should be make more clear what participants did in the two experiments. If the same group of participants was used throughout, does that then mean that some judgments were made more than one or were some judgements reused? E.g., both experiment 1 and experiemnt 2 contain a rating task. Were the selected data in experiment 1 rated again or not?

Minor issues / typo's

P17 (Abstract): Lexical tones and emotions are conveyed by a similar set of acoustic parameters; -> partly similar set (because both tones and emotions are also conveyed by parameters beyond F0)

P18 conveys not only -> not only conveys

P18 Emotional tone is defined as the vocal expression of emotion, which conveys a speaker’s affective states; -> this is a very strong statement given the discussion in the field about the status of emotion as a category and what it exactly is that vocal expression expresses. So, iether more than one reference is needed, or a more nuanced statement (preferably both)

P18 When discussing Scherer's (2003) review, the high correlation between F0 and, particularly, amplitude needs to be mentioned. This is important, because using both F0 and Ampllitude simultaneously as variables needs to take this interdependence into account.

P19 A lexical tone language -> a tonal language (?)

P19 When compared to a neutral tone of voice ... a longer duration [3-12, 24-31]: this maybe a succint summary of the literature, but in order for it to be useful, especially given the many mutually exclusive effects (e.g. a sad voice has a narrower or wider or similar F0 range), some summary conclusion or integration is necessary (over and above "there is some consistancy, but also not"). In addition: all these are comparisons of the emotion to neutral, which is severely limiting the findings, right? That should be acknowledged.

P19 Similarly, when the authors say that "The variability, however, is consistent with the idea that emotion is sociocultural in nature", they are not wrong, but this statement does not connect that well with using basic emotions as a starting point. It would -to some extent- be in line with the dialect theory of emotion (e.g., Elfenbein and Ambady, 2002), which is based on basic emotion theory. So, some more clarity about the theoretical bachground of the authors and this paper is needed. What is the conceptual background here?

P21 In contrast, duration varied significantly among the emotions for Mandarin but not for Italian. > Not a big issue, but this is most likley also partly due to the fact that Italian is a syllable timed language, where there is much less room for variation in syllable duration.

P22 -> In which language was the study by Mullennix et al.?

P23 -> I have a hard time seeing why the study of Nygaard and Queen. is important here: the effects seemed to be semantic, but how does this connect to the tone/emotion tradeoff that is expected by the authors?

P23 / P24 -> the use of the word "compromised" is somewhat ambiguous; explain how emotion compromised. Similarly for the word asymmetrical: asymmetrical in what way?

P24 Dutch speaking learners -> of Mandarin

P24/25 -> The finding about intermediate learners being better seems not so relevant, unless there is a link with emotion, no?

P25 "Depending on specific Mandarin tones"; this is crucial for this paper (I think) and that authors could explain more why they think different tones are affected differently by (different?) emotions (and how this plays out). If this is not possible given the available information, that should be explained, too, then.

P25 "The way emotions are perceived appears to be language-universal": this contradicts earlier statements about the sociocultural nature of emotions.

P27 -> the juxtaposition of acoustics and perception is a bit odd, I'd use production and perception

P28 -> extracted from the carrier phrase -> in isolation

P29 "Considered the four most common basic emotions" -> perhaps rephrase in light of the comments made above

P29 "Based on the literature review" -> It is important to realize / flag that most (if not all) of the review concerns empirical "just so" findings, without much theoretical underpinning as to why the expected effects are predicted. There is not necessarily something wrong with that, but it is important to consider.

P29/30 -"We also expect emotions to be modulated by specific Mandarin tones and talker sex" -> How? And if it is impossible to say how, explain why that is.

P30 Were participants compensated in any way for their participation?

P32 Were the recordings managed by a (stage) director (see, for example, Banse and Scherer 1996) or were the actors working alone?

P32 Acoustic analysis: indicate how many recordings there were (960, I think)

P32 State the aim of the rating procedure: why was it done?

P32 Explain why NEUTRAL was not rated

P33 name the four acoustic measures analyzed

P34 the Functional Data Analysis -> Functional Data Analysis (drop the determiner)

P35 error bars indicat 95% interval -> the 95% interval

P38 and talker -> there apppears to be an extra space (twice)

P45 The different results for stimuli with and without carrier phrase is reminiscent of work doen on (speaker) normalization. Work by Holger Mitterer and Mathias Sjerps, for example, as well early work by Donald Broadbent (the filter theory)

P48 The NEUTRAL emotion -> NEUTRAL category

P49 in the carrier phrase -> when preceded by a carrier phrase

P49 As indicated, (random) item effects should be incorporated in the analysis

P50 "we focus on two interactions" -> but the second one (tone-context) is not really statistically analyzed or introduced (while the other one is).

References

Elfenbein, H. A., & Ambady, N. (2002). On the universality and cultural specificity of emotion recognition: a meta-analysis. Psychological bulletin, 128(2), 203.

Judd, C. M., Westfall, J., & Kenny, D. A. (2012). Treating stimuli as a random factor in social psychology: a new and comprehensive solution to a pervasive but largely ignored problem. Journal of personality and social psychology, 103(1), 54.

Winter, B., & Grice, M. (2021). Independence and generalizability in linguistics. Linguistics, 59(5), 1251-1277.

Laukka, P., Juslin, P., & Bresin, R. (2005). A dimensional approach to vocal expression of emotion. Cognition & Emotion, 19(5), 633-653.

7. PLOS authors have the option to publish the peer review history of their article (what does this mean?). If published, this will include your full peer review and any attached files.

Reviewer #3: **Yes: **Chong Chee Seng

Reviewer #4: **Yes: **Martijn Goudbeek

---

## [Author Response · Author response to Decision Letter 1]

5 Dec 2022

Reviewer #3: In line with the previous two reviewer’s general comments, I agree that the current work is relevant and can be a valuable addition to our current knowledge base.

A: We thank Dr. Chong for recognizing the contribution of this study. 

1. Is the manuscript technically sound, and do the data support the conclusions?

A) Reviewers 1 and 2 both made similar remarks about the disconnect between the broader motivations of the study under review and how it is positioned within the context of similar research. Reviewer 2 recommended that the authors use the work of Liang and Chen (2019) as an anchor point for the current manuscript.

The authors have done well to incorporate the reviewers comments and the Liang and Chen (2019) study is given prominence in the introduction.

However, given the importance of the study as the cornerstone of the current work, this manuscript will benefit from a revision to provide greater clarity and details of the Liang and Chen study. In its current form, I feel that a reader will need to first review Liang and Chen’s thesis in order to follow this manuscript.

A: As instructed, we have provided more details of Liang and Chen’s study in the introduction (lines 201-216). 

B) Reviewer 1 asked why the authors adopted a four basic emotion model (angry, happy, fear and sad) instead of a six emotion model (which on top of the four aforementioned emotion types, also include surprise and disgust).

The authors replied that the decision was made on the basis that most studies on Mandarin have adopted the four emotion model, hence using the four emotion model would make comparisons with previous studies more straightforward.

I am not sure if I quite follow the logic of this reasoning. As the four emotion model is a subset of the six emotion model, straightforward comparisons can still be made if the current study adopted the six emotion model.

Moreover, using the six emotion model has a number of other benefits. One, it would allow comparisons against a much larger literature base (studies using either four or six emotion models) and potentially against the wider literature of other tonal and non-tonal languages.

The second benefit pertains to Reviewer 2’s comment about the inclusion of Neutral as a possible response option for Experiment 2 to reduce chance level accuracy (from 25% to 20%). A six emotion model would achieve a similar effect by having two additional emotion types as response options.

A third advantage, which I feel is more critical, relates to the aim of study. Fundamental Frequency (F0) is considered to be one of the most salient carriers of emotion information as it is argued that F0 may be an acoustic correlate of arousal (Scherer, 2003). For example, emotions with high arousal such as anger may be expressed with a higher mean F0 and conversely, emotions with low arousal such as sadness may be expressed with a lower mean F0. In this regard, of the four emotions examined in this study, only one (Sad) is considered to be of low arousal and the study may therefore benefit from the inclusion of a larger range of emotion types of varying arousal levels to allow a more nuanced study of the interaction between emotions and lexical tone production.

A: We thank Dr. Chong for pointing out the benefits of adopting the six-emotion model. Since the study has been implemented with four-emotion speech materials, we believe the best we can do at this point, without re-running the entire the study, is to offer a broader theoretical foundation and acknowledge the limitation of the four-emotion model. To these ends, we have added a description of theories of emotion in the introduction (lines 70-93). We have also added a description in the conclusion to acknowledge the limitation of the four-emotion model (lines 783-791).

C) Reviewer 2 raised a question regarding the randomisation procedure that was undertaken within each experimental block in Experiment 2. Similarly, I have a concern regarding randomisation but of the experimental blocks. From my reading, I understand that the order of block presentation was fixed for all participants such that they always viewed stimuli in the isolation condition before the context condition.

I am curious about the authors reasoning for this. I feel that this is may be a weakness as confounds such as training or exposure may have an unwanted impact on the observed results (e.g., participants have familiarised themselves with the stimuli by the time they get to the context condition, hence resulting in higher identification accuracy). This is given greater scrutiny especially since the authors are making claims based on the comparison of the isolation to the carrier condition.

A: Dr. Chong is correct that the isolation condition was always presented before the context condition. We agree that familiarization from the isolation blocks could have boosted accuracy in the context blocks. In hindsight, we could have randomized the order of the blocks to mitigate this concern. We have acknowledged this limitation in the conclusion (lines 803-805). We have also added a paragraph in the introduction regarding the benefit of context (lines 266-292). 

D) Reviewer 2 asked why Neutral stimuli were not included in the emotion perception task as the exclusion of this option had raised chance level accuracy rates from 20% to 25%.

The authors responded on page 33, third sentence from the top of the page (pardon the clunky referencing but there doesn’t seem to be line numbering on the revised version), that ‘Neutral was regrettably not included’. I think it would be more meaningful to justify why Neutral was not included, rather than an admission that it was not there.

Nevertheless, in my opinion, the exclusion of Neutral has the benefit of making this a forced choice task that prevented the potential abuse of Neutral as a bin for stimuli that participants were uncertain about.

However, my concern is about why Neutral as a response option was excluded when Neutral stimuli were presented to the participants in the emotion identification task. From my reading, it appears that Neutral stimuli were presented in every experimental block (each experimental block consists of 120 stimuli, 3 syllables, 4 tones, 5 emotions, and 2 repetitions – bottom of pg. 31). Does this also mean that the participants would be giving an incorrect response to the Neutral stimuli by default simply because the Neutral option was not available?

A: We apologize for not including line numbers in the previous revision. We have added line numbers to this revision. 

We thank Dr. Chong for pointing out the benefit of excluding the NEUTRAL response option. In fact, NEUTRAL stimuli were also excluded from the emotion recognition task—we apologize for not making this explicit. That is, in the emotion recognition task, the NEUTRAL stimuli were excluded, resulting in 192 stimuli (3 syllables, 4 tones, 4 emotions, 2 repetitions, and 2 talkers) per block (isolation block and 1 context block). In the tone identification task, all five emotions were included, resulting in 240 stimuli (3 syllables, 4 tones, 5 emotions, 2 repetitions, and 2 talkers) per block. We have revised the description to clarify the design (line 569-574).

E) Both reviewers had raised issues regarding the unclear rationale behind the isolation vs. context comparisons, and the lack of clarity in the discussion of the findings.

Unfortunately, despite the revision, I am unsure if I follow the logic and significance of the comparison between the isolation and context conditions. There were no predictions or references given that would contradict the rather intuitive hypothesis and critical discussions regarding these findings still seems a little thin.

This section could potentially benefit from another rewrite to clearly state the significance of the isolation vs. context comparison, the underlying or competing mechanisms at work, and how this comparison adds to our current knowledge base. In particular, I find the second paragraph of page 29 hard to follow.

A: We agree that the benefit of context is rather intuitive, therefore we did not elaborate on the rationale behind the comparison in the introduction. We did present predictions regarding the comparison in Experiment 2 immediately before the method (lines 520-546). To highlight the rationale for this comparison, we have added a section entitled “Benefit of context” to the introduction. We have also moved the predictions currently in Experiment 2 to this new section.

F) Another point that is unclear to me is the motivation behind the authors’ aim to examine sex differences. From my reading, there is only one justification given, which is to extend Liang and Chen’s (2019) study where all stimuli were produced by a single female speaker (this information was not given in the current manuscript).

Despite the focus on sex differences, it is unclear whether the analyses here were meant to be exploratory or confirmatory. The manuscript does not provide adequate references or predictions on precisely how sex may have an impact on lexical and emotion tone production and why it is important that we study this.

I note that this issue was not raised by the previous reviewers. My reason for bringing this up is that, the manuscript will be easier to read if clearer objectives and deeper insights into the authors’ thought process is given. Moreover, this has implications regarding the appropriateness of the statistical tests undertaken by the authors.

From my understanding, at the broadest level, this study aims to address how emotion (not emotion and sex) affects the production and perception of Mandarin lexical tones. While it is interesting to tease apart and comment on sex differences, I feel that the logical next step, which is missing, is how these effects may be controlled for in order to draw inferences regarding the generalisability of the findings and its robustness against sex and other potential confounds arising from idiosyncratic individual differences.

In this regard, I highly recommend the use of mixed effects models where speaker, sex and perhaps even syllable type may be entered as random intercepts and slopes. This is aligned with Reviewer 2’s suggestion that mixed effects models be applied on data from Experiment 2. I am of the opinion that this should also be applied to Experiment 1 and with more rigour than what is currently done (more on this in the next section).

A: We agree that this study focused on the effect of emotion (not emotion and talker sex) on tone production and perception. As suggested, we have rerun the statistical analysis for Experiment 1 (acoustic analysis) using linear mixed-effects models with talker, talker sex, syllable type, and repetition as random effects.

2. Has the statistical analysis been performed appropriately and rigorously?

A. Reviewer 2 recommended the use of Functional Data Analysis for the analysis of F0 contours.

The authors agreed that the use of Functional Data Analysis was appropriate but unwarranted as the current analysis of contours was intended to be a manipulation check of sorts to show that the four tones were produced as intended.

I agree that the use of Functional Data Analysis may not be necessary if the analysis was intended to be a simple visual sanity check. However, in its current form, explorations of the contours appear to be beyond the level of a simple sanity check as contrasts between sex and emotions were made. It is therefore inaccurate for the authors to claim that the ‘qualitative observations were meant to corroborate that the tones were ‘produced as intended’ (second last sentence of first paragraph on page 19) as it is not clear what the ‘intended’ shape of the contour is for the different sexes and emotion types.

My recommendation is for the authors to acknowledge that this is an exploratory examination to visualise and better understand the stimuli produced in the current study, rather than framing this as a comparison against a ‘standard’.

A: We agree that the F0 plot was intended to show how the Mandarin tones were produced by talkers in this study. As suggested, we have added a note about this point and deleted statements about potential sex and emotion contrasts.

B. Reviewer 2 suggest that that mixed-model multinomial regression analysis in R may be a more appropriate analysis than the repeated measures ANOVA performed by the authors.

The authors have now applied mixed model logistic regression in their analysis of the data from Experiment 2.

More critically, I am concerned about how the variables were entered into the model in the revised manuscript. I am not sure if I understand why participants were entered as a random effect instead of variables such as sex, talker or syllable. It also unclear why the model was needlessly overcomplicated by the inclusion of the interaction effect between emotion and context when this is not the aim of the study (and also subsequently ignored by the authors – pg. 39). It would also be instructive for the authors to cite the R package that was used.

The significant interactions effects were further illustrated by the authors through the use of confusion matrices. While the findings presented are sound, I feel that an edit would provide better clarity. For example, the authors wrote ‘The most common error for Mandarin tones produced with ANGRY was Tone 4’. My suggestions would be to rephrase this to something along the lines of ‘There appears to be a response bias where Mandarin lexical tones, Tones 1 and 3 in particular, were most commonly misidentified as Tone 4 when presented in an Angry tone of voice’.

A: We thank the reviewer for the suggestion to clarify and simplify the regression model. As suggested, we have rerun the analysis by focusing on factors most relevant to our research questions. In particular, fixed effects now include tone, emotion, context, and tone-emotion interaction. Random effects now include talker, syllable type, and repetition. 

The R packages we used included lme4 and car. As instructed, we have included the following citations:

Bates D, Mächler M, Bolker B, Walker S (2015). “Fitting Linear Mixed-Effects Models Using lme4.” Journal of Statistical Software, 67(1), 1–48. doi:10.18637/jss.v067.i01.

Fox J, Weisberg S (2019). An R Companion to Applied Regression, Third edition. Sage, Thousand Oaks CA. https://socialsciences.mcmaster.ca/jfox/Books/Companion/.

As for the confusion matrices, we have also revised the descriptions as the reviewer suggested to improve the clarity of our interpretation.

C. I would also like the authors to review how the confidence interval error bars of Figures 3 and 6 were generated. Some of the CI bars are in the negative range, which although can happen through calculation but not in actual measure (e.g., F0 range). This is likely indicative of a rather unusual or problematic distribution in the data which calls into question whether sufficient data preparation and cleaning (outliers?) has been conducted prior to analysis.

A: Thank you for the observation. We have replaced the original figures with boxplots, which show the data distribution more clearly.

D. When conducting regression analysis, it is good practice to include within the body of the results section the statistics of the analysis such as the Beta estimates, standard errors, and Z-ratio. These are currently only available as downloadable supplementary information.

A: We thank Dr. Chong for this suggestion. Since the manuscript is already quite long, we feel that including detailed statistics in main text will make reading difficult. Therefore we have chosen to keep the details in the supplementary information.

3. Have the authors made all data underlying the findings in their manuscript fully available?

A. Following the provided link, I am able to view only a simplified output of the mixed effects model. No other data was found.

A: Thank you for this observation. We included the full output of a linear mixed-effect logistic regression model in the supporting information.

4. Is the manuscript presented in intelligible fashion and written in standard English?

A. In general, both reviewers commented to the effect that edits were required to improve the quality of the manuscript. Reviewer 2 further noted that the authors frequently made unsubstantiated claims.

The authors have made extensive rewrites with assistance from a native speaker of English who provided feedback on word choice, grammar and writing style.

It is clear that the quality of the writing has improved in the revised manuscript on account of the substantial effort put in by the authors’ and their openness to suggestions. However, it can definitely benefit from another round of polishing and editing as there are areas within the manuscript (e.g., the section on confusion matrices) that although is grammatically sound, lacks clarity.

A: As suggested, we have revised section on confusion matrices to improve clarity. 

Reviewer #4: 

This manuscript addresses and interesting question, namely how linguist and paralinguistic cues mutually influence each other in communication. Specifically, it does this by investigating how the expression of emotion in Mandarin affects the production and perception of tones 1 to 4 and, conversely, how the production of tones 1 to 4 affects the production and perception of emotion. A tonal language such as Mandarin provides an excellent testbed for the linguistics/paralinguistics interface, because emotional expression often modulates pitch, and pitch is one of the defining features (if not the defining feature) of Mandarin tones.

Since I did not review the original manuscript, I tried to pay close attention to the original comments and the replies from the authors. However, even though I did take the original reviews as a basis, there are some new remarks and recommendations in my review.

Theoretical points

------------------

One of the points of discussion in the original reviews was how to defend the use of four emotion categories. The authors have explained that these four are the four basic emotions in Ortony and Turners 1990 paper. This is somewhat unfortunate, since this paper is a highly cited /critique/ of the whole idea of basic emotion theory (a critique deemed relevant enough for Paul Ekman to directly engage the paper in his 1992 paper). That said, and although I would appreciate a better motivation for the -small number of- emotions included in the corpus, I do not particularly mind that there are only four. The more fundamental problem is that the low number of emotions (and the ones chosen) are never discussed in terms of the consequences for recognition in the vocal domain. For example, the fact that there is only one positive emotion (happiness) and only one low aroused emotion (sadness) strongly influences the decision problem that participants face. Likewise, it also strongly influences the results of the analysis, since (for example) the acoustic profile of sadness is so much different from the others in intensity, that significant results are bound to emerge. Contrast this with a situation where other low aroused emotions such as disgust or tenderness are in the dataset and you get very different results for, say, F0 or intensity. For a balanced and transparent discussion of the meaning of the results, factors like these needs to be taken into account, both in introducing the study and in the discussion section.

Along these lines I think the exclusive focus on basic emotions is needlessly limiting. Especially considering the fact that results and conclusions are often cast in terms of positive versus negative emotions, I think the dimensional approach to emotions (e.g., Russel, 1980, Russell and Feldman Barrett, 1999 and further) should be given proper attention (for example, when describing the emotions in the study, but also when interpreting the results, which is, as mentioned often already done in terms of valence -which is notably not a property of basic emotion theory which considers all emotions orthogonal categories). A reference to Laukka 2005 (in the rebuttal letter) is somewhat misleading, since although it provides evidence for categorical (which is not the same) perception of emotions, a paper from the same author in the same year using the same dataset uses the dimensional approach (Laukka, Juslin, & Bresin, 2005).

A: We thank Dr. Goudbeek for this comment. Dr. Chong also raised a similar comment. As noted in our earlier response, we have added a description of theories of emotion in the introduction (lines 70-93). We have also added a description in the conclusion to acknowledge the limitation of the four-emotion model (lines 783-791).

Methodolical / Statistical points

In addition to these theoretical considerations, there are some methodological issues that need to be addressed.

For the analysis of variance in experiment 1, the (statistical) design is somewhat unclear. The design is introduced with "For each of the four acoustic measures, a three-way mixed-design analysis of variance (ANOVA) was conducted [...] with emotion [] and Mandarin tones [...] as within-subject factors and talker sex [...] as a between-subject factor." What is not clear is why tone and emotion are within factors, but why other stimulus characteristics that were deemed relevant in the construction of the corpus (syllable, repetition) were not used as within factors. Statistically, this variance is unexplained variance, but in a more sophisticated analysis, these could be random effects over which the study could generalize (see Judd, Westfall, and Kenny, 2012, but also Winter and Grice, 2021). In any case, the precise number of items in the analysis should be clarified, if only because otherwise the degrees of freedom in the analysis become difficult to interpret.

A: We thank Dr. Goudbeek for this observation. As noted in our response to Dr. Chong, we have rerun the statistical analysis using linear mixed-effects models with talker, talker sex, syllable type, and repetition as random effects.

As a follow up to the same analysis (the mixed ANOVA of experiment 1), the authors use LSD as a post hoc test. This test does not correct in any way for multiple comparisons, thus increasing the possibility of a type I error. While there might be arguments to not correcting every multiple comparison with a bonferroni test, the choice to not correct at all (certainly with such a large number of comparisons. This is particularly relevant in light of Figure 1, where almost all CI's overlap substantially (indicating the lack of a significant difference) which is in stark contrast to the findings of the post hoc analysis.

A: We agree with Dr. Goudbeek about the need to control for type I error. As suggested, we have used the Tukey test in the revised statistical analysis to correct for multiple comparisons.

For the second analysis, multinomial mixed effects models are indeed the correct way to analyze this kind of data. However, mixed models also enable the inclusion of items (in addition to participants) as a random effect (again, see Judd, Westfall, and Kenny, 2012, but many others). This appears to not have been done, severely limiting the generalizability of the findings. For the revision, items should be entered as a random effect in the analysis.

A: Thank you for this observation. Dr. Chong also made a similar point. As suggested, we have included item characteristics (talker, syllable type, and repetition) as random effects.

It should be make more clear what participants did in the two experiments. If the same group of participants was used throughout, does that then mean that some judgments were made more than one or were some judgements reused? E.g., both experiment 1 and experiment 2 contain a rating task. Were the selected data in experiment 1 rated again or not?

A: The raters for Experiment 1 were different from the listeners for Experiment 2. Therefore no judgments were reused. We apologize for the confusion.

Minor issues / typo's

P17 (Abstract): Lexical tones and emotions are conveyed by a similar set of acoustic parameters; -> partly similar set (because both tones and emotions are also conveyed by parameters beyond F0)

A: Corrected.

P18 conveys not only -> not only conveys

A: Corrected.

P18 Emotional tone is defined as the vocal expression of emotion, which conveys a speaker’s affective states; -> this is a very strong statement given the discussion in the field about the status of emotion as a category and what it exactly is that vocal expression expresses. So, either more than one reference is needed, or a more nuanced statement (preferably both)

A: We have revised this statement and added more references (lines 49-50).

P18 When discussing Scherer's (2003) review, the high correlation between F0 and, particularly, amplitude needs to be mentioned. This is important, because using both F0 and Amplitude simultaneously as variables needs to take this interdependence into account.

A: We have added information about the correlation as suggested (line 54-55).

P19 A lexical tone language -> a tonal language (?)

A: Revised.

LN71 When compared to a neutral tone of voice … a longer duration [3-12, 24-31]: this maybe a succinct summary of the literature, but in order for it to be useful, especially given the many mutually exclusive effects (e.g. a sad voice has a narrower or wider or similar F0 range), some summary conclusion or integration is necessary (over and above “there is some consistency, but also not”). In addition: all these are comparisons of the emotion to neutral, which is severely limiting the findings, right? That should be acknowledged.

A: We suspect the conflicting findings arise from methodological differences between studies. We have revised the summary to elaborate on this point (lines 96-106 & lines 115-120). 

We understand that comparing emotions to each other would be quite informative. Our intention was to use neutral as a baseline to keep the number of comparisons manageable. We have acknowledged this limitation in the revised manuscript (lines 803-805). 

P19 Similarly, when the authors say that "The variability, however, is consistent with the idea that emotion is sociocultural in nature", they are not wrong, but this statement does not connect that well with using basic emotions as a starting point. It would -to some extent- be in line with the dialect theory of emotion (e.g., Elfenbein and Ambady, 2002), which is based on basic emotion theory. So, some more clarity about the theoretical background of the authors and this paper is needed. What is the conceptual background here?

A: We thank Dr. Goudbeek for this observation and the reference. We have included the reference to support the statement. Since the focus of this study is on how emotions affect the acoustics and perception of Mandarin tones, rather than the effect of culture on emotion expression, we believe this fascinating topic can be addressed in a different study.

P21 In contrast, duration varied significantly among the emotions for Mandarin but not for Italian. > Not a big issue, but this is most likely also partly due to the fact that Italian is a syllable timed language, where there is much less room for variation in syllable duration.

A: Thank you for this observation. We have added this point to the text.

P22 -> In which language was the study by Mullennix et al.?

A: It was English. We have clarified this.

P23 -> I have a hard time seeing why the study of Nygaard and Queen. is important here: the effects seemed to be semantic, but how does this connect to the tone/emotion tradeoff that is expected by the authors?

A: We agree that this study is not super relevant to the issue of tone-emotion relationship. However, the point of this section is to show that emotion can affect spoken word recognition, which necessarily involves processing meaning. If we removed this reference, then all studies of word recognition reviewed in this section should be removed. With these considerations, we would like to keep this reference.

P23 / P24 -> the use of the word "compromised" is somewhat ambiguous; explain how emotion compromised. Similarly for the word asymmetrical: asymmetrical in what way?

A: We have replaced these words with a more elaborated description.

P24 Dutch speaking learners -> of Mandarin

A: Corrected.

P24/25 -> The finding about intermediate learners being better seems not so relevant, unless there is a link with emotion, no?

A: This finding is based on proficiency-tone interaction and the effect does not involve emotion. We agree that this does not seem relevant, so we have removed this statement.

P25 "Depending on specific Mandarin tones"; this is crucial for this paper (I think) and that authors could explain more why they think different tones are affected differently by (different?) emotions (and how this plays out). If this is not possible given the available information, that should be explained, too, then.

A: Thank you for the observation. The tone-emotion interaction was indeed a major finding of the study. We did not have sufficient information from the literature to make predictions about which tones would be affected more by emotions. Our data also did not reveal consistent tone-specific patterns to allow meaningful speculations beyond the presence of the interaction itself.

P25 "The way emotions are perceived appears to be language-universal": this contradicts earlier statements about the sociocultural nature of emotions.

A: Thank you for pointing this out. This statement has been deleted.

P27 -> the juxtaposition of acoustics and perception is a bit odd, I'd use production and perception

A: We still think acoustics is a more precise term that describes what we did in this study, i.e., acoustic analysis of speech produced by the talkers. Speech production could be examined with physiological measures, but that is not what we did in this study.

P28 -> extracted from the carrier phrase -> in isolation

A: We had used “in isolation” in the original version, but one of the original reviewers asked us to clarify that the isolated syllables were extracted from the carrier phrase, not produced in isolation. We agree that “in isolation” is a more succinct term and have used it in this revision.

P29 "Considered the four most common basic emotions" -> perhaps rephrase in light of the comments made above

A: We have rephrased this statement to highlight different perspectives on characterizing emotions.

P29 "Based on the literature review" -> It is important to realize / flag that most (if not all) of the review concerns empirical "just so" findings, without much theoretical underpinning as to why the expected effects are predicted. There is not necessarily something wrong with that, but it is important to consider.

A: Thank you for the observation. We acknowledge that we did not have sufficient information to make predictions about specific tones or to interpret the tone-emotion interactions fully.

P29/30 -"We also expect emotions to be modulated by specific Mandarin tones and talker sex" -> How? And if it is impossible to say how, explain why that is.

A: At Dr. Chong’s suggestion, we have removed talker sex as a fixed effect. 

P30 Were participants compensated in any way for their participation?

A: Yes, all participants were compensated. We have included payment information in the revised manuscript. 

P32 Were the recordings managed by a (stage) director (see, for example, Banse and Scherer 1996) or were the actors working alone?

A: Before the recording, the first author discussed with the actors the emotional tones that they should aim for. The actors completed the recording in a sound-treated booth while the first author monitored the recording. We have added this information to the revised manuscript.

P32 Acoustic analysis: indicate how many recordings there were (960, I think)

A: Yes, there were 960 (3 syllables * 4 tones * 5 emotions * 2 repetitions * 8 speakers) speech samples in Experiment 1. We have added this information to the revised manuscript.

P32 State the aim of the rating procedure: why was it done?

A: The rating was done to make sure the intended emotions were actually present in the speech materials. This was explained in the section on acoustic analysis.

P32 Explain why NEUTRAL was not rated

A: We did not think neutral stimuli needed to be rated because they were emotionally neutral. In hindsight, we agree that it would have been a good idea to include neutral stimuli in the rating task too.

P33 name the four acoustic measures analyzed

A: The names of the four acoustic measures are specified in lines 402-404.

P34 the Functional Data Analysis -> Functional Data Analysis (drop the determiner)

A: Corrected.

P35 error bars indicate 95% interval -> the 95% interval

A: Corrected.

P38 and talker -> there appears to be an extra space (twice)

A: Corrected. 

P45 The different results for stimuli with and without carrier phrase is reminiscent of work done on (speaker) normalization. Work by Holger Mitterer and Mathias Sjerps, for example, as well early work by Donald Broadbent (the filter theory)

A: Thank you for making the connection with talker normalization. We have included this idea in the section on the benefit of context in the introduction.

P48 The NEUTRAL emotion -> NEUTRAL category

A: Thank you for the suggestion. By “NEUTRAL emotion” we meant NEUTRAL as an emotional tone of voice (defined in lines 46). Since Mandarin has a neutral (lexical) tone, we feel that labeling NEUTRAL as an emotion avoids the potential confusion with the lexical tone.

P49 in the carrier phrase -> when preceded by a carrier phrase

A: We have changed this phrase when it first appears in line 291-292. 

P49 As indicated, (random) item effects should be incorporated in the analysis

A: Yes, as noted earlier, we have included item characteristics as random effects.

P50 "we focus on two interactions" -> but the second one (tone-context) is not really statistically analyzed or introduced (while the other one is).

A: In the revised modeling, we examined only the tone-emotion interaction in light of Dr. Chong’s suggestion. Therefore we are no longer discussing the tone-context interaction.

---

## [Decision Letter · Decision Letter 2]

16 Feb 2023

PONE-D-21-23777R2Emotional tones of voice affect the acoustics and perception of Mandarin tonesPLOS ONE

Dear Dr. Chu,

Thank you for submitting your manuscript to PLOS ONE. First, let me apologize for the lengthy review process. I notice that the manuscript has been submitted for more than 500 days and has undergone 2 rounds of major revision. When I took up the responsibility in handling this manuscript, my goal was to ensure that you do not need to undergo another round of major review. Therefore, I tried my best to engage previous reviewers. Unfortunately, they were unavailable. Given that in the second round of review, only one reviewer requested for Major Revision, I decided to invite one new reviewer only to speed up the process. I have also reviewed your manuscript to give you additional comments.

Both the new reviewer and I agree that the manuscript is potentially publishable. However, there are some minor flaws that need to be corrected before I can accept it for publication. Therefore, I am making the decision of Minor Review. Please check the manuscript carefully and consider seeking help from copyediting services.

You can find my comments below (under Additional Editor Comments). You can also find the comments of the additional reviewer in this decision letter. Please address them in your revision.

We look forward to receiving your revised manuscript.

Kind regards,

Yiu-Kei Tsang

Academic Editor

PLOS ONE

Journal Requirements:

Additional Editor Comments:

Personally, I think how emotional tones and lexical tones interacted in the perception of Mandarin is a very interesting topic. While I agree with the previous reviewers that the experimental design is not perfect (e.g., not having all six emotions and not having a neutral response option), I believe the authors have discussed their results in an unbiased manner by stating the limitations explicitly. Given that no experiment is perfect, I think the most important point is to provide enough information so that readers can make informed judgement about the study and be inspired to conduct better experiments to clarify uncertainties. With this in mind, I am willing to support the manuscript for publication.

Reviewers' comments:

Reviewer's Responses to Questions

**Comments to the Author**

1. If the authors have adequately addressed your comments raised in a previous round of review and you feel that this manuscript is now acceptable for publication, you may indicate that here to bypass the “Comments to the Author” section, enter your conflict of interest statement in the “Confidential to Editor” section, and submit your "Accept" recommendation.

Reviewer #5: (No Response)

2. Is the manuscript technically sound, and do the data support the conclusions?

Reviewer #5: Yes

3. Has the statistical analysis been performed appropriately and rigorously? 

Reviewer #5: Yes

4. Have the authors made all data underlying the findings in their manuscript fully available?

Reviewer #5: No

5. Is the manuscript presented in an intelligible fashion and written in standard English?

Reviewer #5: Yes

6. Review Comments to the Author

Reviewer #5: I generally agree with previous review comments regarding the rationale behind and the design of the study (e.g., the four-emotion model, lack of counterbalancing on the conditions). I think the authors have adequately addressed these concerns in the last round of revisions, by providing additional justification or acknowledging the limitations. But I still noticed quite a few inaccuracies in both the results and the writing, and I hope the authors can do another round of proofreading and editing. I think this request has been raised repeatedly in previous rounds of review, and the authors should take the suggestion more seriously. Below I list some of these inaccuracies together with other minor comments. (I say “some” because I might not have spotted all of them.)

Line 352, 557: How much is the compensation in USD?

Line 368: “a total of 960 stimuli for each participant”, delete “for each participant”?

Line 663: “accuracy appears higher for target syllables presented than in isolation”, presented in context?

Line 673: “and the tone-emotion interaction fixed effects”, as fixed effects?

Line 679: “comparisons were indicates that”?

Line 702: Not all common errors are highlighted, e.g., Tone 1 SAD identified as HAPPY for 25%.

Line 705: “ranging from 86% to 99%”, ranging from 85% to 99%?

Line 781: “but it affects the identification of specific Mandarin tones to a greater extent than it affects the recognition of specific emotions”, this is not fully correct considering that the context seemed to boost emotion recognition accuracy to a larger extent (from 21%-93% to 85%-99%; tone identification from 40%-98% to 78%-100%)?

Figure 5: “Duration (ms)”, Duration (s)?

Figures 6, 7: Better specify tone identification /emotion recognition accuracy rather than “Accuracy”, in the figures.

7. PLOS authors have the option to publish the peer review history of their article (what does this mean?). If published, this will include your full peer review and any attached files.

Reviewer #5: No

---

## [Author Response · Author response to Decision Letter 2]

13 Mar 2023

Additional Editor Comments

Personally, I think how emotional tones and lexical tones interacted in the perception of Mandarin is a very interesting topic. While I agree with the previous reviewers that the experimental design is not perfect (e.g., not having all six emotions and not having a neutral response option), I believe the authors have discussed their results in an unbiased manner by stating the limitations explicitly. Given that no experiment is perfect, I think the most important point is to provide enough information so that readers can make informed judgement about the study and be inspired to conduct better experiments to clarify uncertainties. With this in mind, I am willing to support the manuscript for publication.

Thank you for recognizing the potential contribution of this study. We agree that there is always room for improvement. Thank you for supporting this manuscript for publication. 

Comments to the Author

Reviewer #5: I generally agree with previous review comments regarding the rationale behind and the design of the study (e.g., the four-emotion model, lack of counterbalancing on the conditions). I think the authors have adequately addressed these concerns in the last round of revisions, by providing additional justification or acknowledging the limitations. But I still noticed quite a few inaccuracies in both the results and the writing, and I hope the authors can do another round of proofreading and editing. I think this request has been raised repeatedly in previous rounds of review, and the authors should take the suggestion more seriously. Below I list some of these inaccuracies together with other minor comments. (I say “some” because I might not have spotted all of them.)

We appreciate the reviewer’s detailed and thoughtful comments. We have made corrections based on the reviewer’s suggestions. We have also invited a colleague (a native speaker of English) to proofread and edit the manuscript.

1. Line 352, 557: How much is the compensation in USD?

We have converted the currency to USD: $54 (LN352) and $34 (LN557) USD.

2. Line 368: “a total of 960 stimuli for each participant”, delete “for each participant”?

Thank you for noting the redundancy. We have deleted it.

3. Line 663: “accuracy appears higher for target syllables presented than in isolation”, presented in context? 

Yes, we have revised it to “accuracy appears higher for target syllables presented in context than in isolation.” Thank you for catching the missing phrase.

4. Line 673: “and the tone-emotion interaction fixed effects”, as fixed effects? 

Yes, we have added “as before “fixed effects”. Thank you.

5. Line 679: “comparisons were indicates that”? 

We have revised it to “comparisons indicate that.”

6. Line 702: Not all common errors are highlighted, e.g., Tone 1 SAD identified as HAPPY for 25%.

Our intention was to highlight the most common error that exceed 20%, not all errors exceeding 20%. Please see caption (LN 700-701). In this case, it is FEAR at 29%. 

7. Line 705: “ranging from 86% to 99%”, ranging from 85% to 99%? 

Thank you. We have updated the percentage.

8. Line 781: “but it affects the identification of specific Mandarin tones to a greater extent than it affects the recognition of specific emotions”, this is not fully correct considering that the context seemed to boost emotion recognition accuracy to a larger extent (from 21%-93% to 85%-99%; tone identification from 40%-98% to 78%-100%)? 

Thank you for the observation. We have deleted this clause. 

9. Figure 5: “Duration (ms)”, Duration (s)? 

Thank you. We have corrected the label.

10. Figures 6, 7: Better specify tone identification /emotion recognition accuracy rather than “Accuracy”, in the figures.

Thank you. We made the changes as suggested.

---

## [Editor Report · Decision Letter 3]

14 Mar 2023

Emotional tones of voice affect the acoustics and perception of Mandarin tones

PONE-D-21-23777R3

Dear Dr. Chu,

We’re pleased to inform you that your manuscript has been judged scientifically suitable for publication and will be formally accepted for publication once it meets all outstanding technical requirements.

Kind regards,

Yiu-Kei Tsang

Academic Editor

PLOS ONE
---

## [Editor Report · Acceptance letter]

27 Mar 2023

PONE-D-21-23777R3 

Emotional tones of voice affect the acoustics and perception of Mandarin tones 

Dear Dr. Chu:

I'm pleased to inform you that your manuscript has been deemed suitable for publication in PLOS ONE. Congratulations! Your manuscript is now with our production department. 

Kind regards, 

on behalf of

Dr. Yiu-Kei Tsang 

Academic Editor

PLOS ONE